# Probing the pathways of free charge generation in organic bulk heterojunction solar cells

Jona Kurpiers[1], Thomas Ferron[2], Steffen Roland[1], Marius Jakoby[3], Tobias Thiede[1], Frank Jaiser[1], Steve Albrecht[4], Silvia Janietz[5], Brian A. Collins[2], Ian A. Howard [3] & Dieter Neher[1]

The fact that organic solar cells perform efficiently despite the low dielectric constant of most photoactive blends initiated a long-standing debate regarding the dominant pathways of free charge formation. Here, we address this issue through the accurate measurement of the activation energy for free charge photogeneration over a wide range of photon energy, using the method of time-delayed collection field. For our prototypical low bandgap polymer:fullerene blends, we find that neither the temperature nor the field dependence of free charge generation depend on the excitation energy, ruling out an appreciable contribution to free charge generation though hot carrier pathways. On the other hand, activation energies are on the order of the room temperature thermal energy for all studied blends. We conclude that charge generation in such devices proceeds through thermalized charge transfer states, and that thermal energy is sufficient to separate most of these states into free charges.

[1] Institute of Physics and Astronomy, Soft Matter Physics, University of Potsdam, Karl-Liebknecht-Str. 24-25, 14476 Potsdam-Golm, Germany. [2] Department of Physics and Astronomy, Washington State University, 100 Dairy Road, Pullman, WA 99164, USA. [3] Karlsruhe Institute of Technology (KIT), Institute of Microstructure Technology (IMT), Hermann-von-Helmholtz Platz-1, 76344 Eggenstein-Leopoldshafen, Germany. [4] Helmholtz-Zentrum Berlin für Materialien und Energie, Nachwuchsgruppe Perowskit Tandemsolarzellen, Kekuléstraße 5, 12489 Berlin, Germany. [5] Fraunhofer IAP, Polymere und Elektronik, Geiselbergstraße 69, 14476 Potsdam-Golm, Germany. Correspondence and requests for materials should be addressed to D.N. (email: neher@uni-potsdam.de)

Recent efforts in materials and device architecture optimization have pushed the power conversion efficiency of bulk heterojunction organic solar cells (BHJ OSCs) well beyond 10%[1–3]. All of these devices comprise at least two semiconductor components: an electron donor and an electron acceptor. Primary photoexcitations in the photoactive blends are strongly bound singlet excitons on either of these two semiconductors; these excitons dissociate at the donor–acceptor (D–A) interface to form interfacial charge transfer (CT) states. These intermolecular excitations will eventually split into free charges or undergo geminate recombination. Because of the low dielectric constant $\varepsilon_r$ being approximately 3.5 of most organic semiconductors, electron-hole pairs in intramolecular excitons and intermolecular CT excitations experience strong Coulombic binding—in apparent contradiction to the efficient and field-independent free charge generation of several state-of-the-art D–A blends. This discrepancy triggered an intensive discussion regarding the pathway of absorbed photon-to-electron conversion and, in particular, the nature and energetics of the CT states primarily involved in free carrier formation[4–7].

It has been proposed that the energy of the primary singlet exciton is partially converted into electronic and/or vibronic excitations of the interfacial CT states and that this excess energy assists in the split-up of the CT states into free carriers, thereby overcoming the mutual Coulomb binding[8–11]. Support for this depiction comes from theoretical and experimental investigations suggesting that exciton dissociation generates hot CT states and that these states are the predominant precursors of free charges. For example, Grancini et al. used transient absorption spectroscopy (TAS) to demonstrate that high energy excitons in the polymer preferably form electronically excited CT states, which split into free charge carriers on the time scale of 100 fs[8]. The strong coupling of the excited singlet excitons and hot CT states was convincingly confirmed by the theoretical calculations of, for example, Troisi et al.[12]. Bakulin et al. showed that excited CT states have a higher probability to split up into free charges, using a pump-push-probe method[13]. This experimental evidence was supported by quantum chemical studies demonstrating the pronounced delocalization of the holes in the excited CT states.

Related to that, results from transient optical measurements suggested an energy derived two-pool model, where dissociation of intramolecular excitons at the D–A heterojunction forms either low energy bound CT excitons, prone to geminate recombination but contributing very little to free charge generation, or more delocalized higher energy bound polaron pairs, with a low probability for geminate recombination but a high efficiency to split into free charges[14,15]. If there is little coupling between these two state manifolds, the efficiency of free charge formation will be determined by the initial branching ratio, meaning that it is decided on a very early time scale, before thermalization is completed, whether exciton dissociation occupies bound or dissociable CT states[16,17].

Other work suggested that free charge formation involves thermalized CT states. Most of these studies utilized that by definition, CT states can be populated directly through low energy (sub-bandgap) excitation. For example, the work by Vandewal and coworkers revealed very similar shapes of the absorption spectra and of the external quantum efficiency (EQE) for free charge formation, indicating that the IQE does not depend on whether the CT state or an exciton is the primary excitation[18]. Our group and others employed time-delayed-collection-field (TDCF) experiments to investigate free carrier formation as a function of electric field and excitation energy for different BHJ blends. These studies revealed there was neither an appreciable effect of the excess energy on the yield of free carrier formation, nor on the field-dependence of the free carrier

generation, thereby supporting the view that charge generation proceeds through thermalized CT states[18–20].

The aim of this communication is to contribute to the current debate regarding the dominant pathway of free charge formation in bulk heterojunction solar cells. To this end, we perform precise measurements of the temperature dependence of free charge formation for a wide range of excitation energies, including direct CT excitation. It is a well proven fact that the activation energy is highly sensitive to the energy landscape involved in generation of free charges[21–24], rendering it as an important parameter when searching for the dominant pathway of photogeneration. For example, the dependence of the activation energy on excitation conditions has been often used to prove hot processes in neat organic semiconductors, where it arises because singlet excitons with higher energy form bound electron-hole pairs with a larger thermalization distance, thereby increasing the likelihood of free charge formation[25–27]. For organic blends, the activation energy was shown to vary significantly, between few tens to hundreds of meV depending on the system and the methodology[15,21,28,29], but we are not aware of a study of the activation energy over a sufficient large range of excitation energy, well below and above the bandgap.

Our method of choice is TDCF which is an optoelectronic method where charges are photogenerated at variable pre-bias and subsequently extracted with a high collection field (Fig. 1a)[30–32]. In contrast to TAS, this method measures free (extractable) charge only, while compared to photocurrent experiments, issues related to poor charge transport properties are insignificant. Combined with the results from sensitive PL and EL experiments we conclude that photoexcitation in our blends populates a low energy (thermalized) CT manifold, independent of photon energy, that this manifold is the precursor of most free charge and CT photon emission, and that the thermal energy present at room temperature is sufficient to render CT dissociation competitive to geminate recombination.

## Results

**Materials selection and methods.** Our system of choice is the D–A copolymer poly[2,6-(4,4-bis-(2-ethylhexyl)-4H-cyclopenta [2,1-b;3,4-b']-dithiophene)-alt-4,7-(2,1,3-benzothiadiazole)] (PCPDTBT) blended with either [6,6]-phenyl C70-butyric acid methyl ester (PCBM) or indene-C60 bisadduct (ICBA) and the monofluorinated derivative of the copolymer, 1F-PCPDTBT[33], blended with PCBM. Several other prototypical polymer-based blends were excluded from this study following the rationale in Supplementary Note 1. PCPDTBT is a low bandgap polymer, allowing for a wide range of excitation energies, and it exhibits well-expressed CT absorption and emission properties in blends with fullerenes[34]. As another benefit, its blend energetics, morphology, and photovoltaic performance can easily be tuned with minor changes of the chemical structure of the blend constituents as outlined below.

A detailed description of the TDCF setup used for the present work has been provided previously[32]. In brief, a femtosecond (or nanosecond) laser pulse creates charges as a device is held at particular biases. After the shortest possible delay of 4 ns, the voltage is ramped up to a high reverse bias to collect all extractable charges present in the layer. TDCF experiments as function of fluence, delay time and temperature, combined with transient photoluminescence experiments revealed that under the chosen conditions most photogenerated species either undergo geminate recombination or dissociate into free carriers within the first few nanoseconds, even for the lowest considered temperature and zero internal field (Supplementary Note 2). Therefore, TDCF experiments performed at a delay time of 4 ns provide an accurate

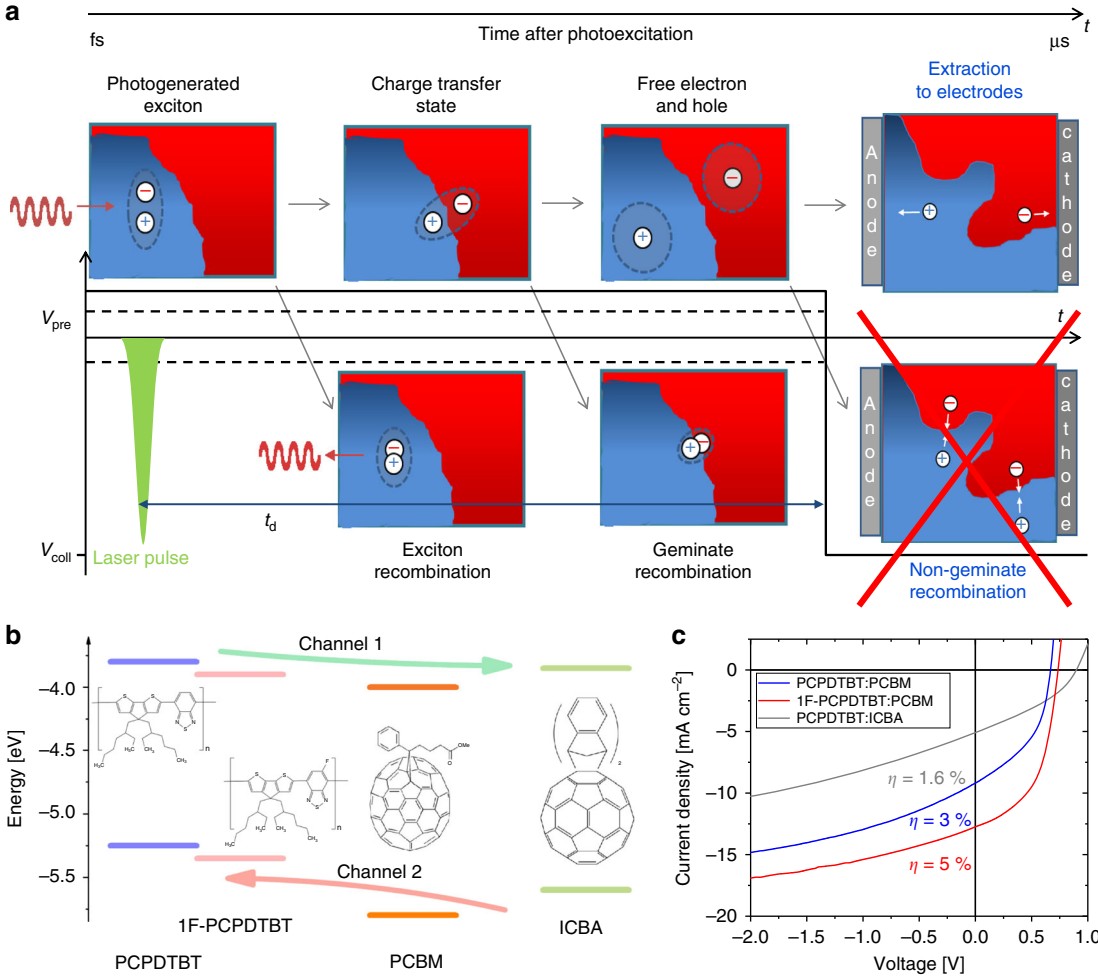

**Fig. 1** Scheme of the TDCF method, energy level diagram and device performances: **a** Scheme of the elementary excitations and processes involved in absorbed photon-to-electron formation. Geminate and non-geminate recombination occur at different time scales at AM1.5G-relevant carrier densities. Therefore, applying a strong reverse bias in TDCF prior to the onset of non-geminate recombination allows free charge formation due to the separation of CT states to be measured. **b** Energy level diagram for the materials used in this study. Arrows indicate the pathways of charge formation via electron (channel 1) or hole transfer (channel 2) across the donor-acceptor heterojunction. The energy offset at the heterojunction is reduced by substituting PCBM with ICBA or PCPDTBT with 1F-PCPDTBT. **c** Current voltage characteristics of the studied blends with a thickness of approx. 110 nm under simulated AM1.5 G illumination. As expected, increasing the energy of the charge separated state (by either lowering the polymer HOMO and/or heightening the acceptor LUMO) increases the $V_{oc}$, but there is no trivial connection between the energetics of the blend and the overall device performance

measurement of the photogenerated free charge density. In all experiments described in the following, the laser fluence was adjusted to generate charge carrier densities that are comparable to those under steady state AM1.5 G illumination at room temperature and that are low enough to avoid non-linear losses during the 4 ns delay time.

**Electronic structure and blend morphology.** The chemical structures of the used compounds, as well as the energetic positions of their lowest unoccupied molecular orbitals (LUMO) and highest occupied molecular orbitals (HOMO) obtained from literature[19,33], are displayed in Fig. 1b. Fluorination lowers the energy of both the HOMO and LUMO by $ca$. 0.1 eV, which will simultaneously reduce the LUMO–LUMO and HOMO–HOMO offsets at the heterojunction. Moreover, as the energy of the polymer singlet exciton is unaffected by fluorination, the lower polymer LUMO will decrease the driving force for charge separation, $\Delta E_{CS}$. Replacing PCBM with ICBA has a similar effect on the overall energetics, with an increase in the energy of the charge separated state by ca. 0.2 eV and a corresponding

reduction in the LUMO–LUMO and HOMO–HOMO offsets. As expected, raising the energy of the charge separated state clearly correlates with an increase in the open circuit voltage (see Fig. 1b, c), while the correlation between the blend energetics and the shape of the JV curve is highly non-trivial, asking for a detailed study of the generation pathways in relation to blend morphology.

Molecular ordering in the studied blends was probed through grazing incident wide angle X-ray scattering (GIWAXS) and Fig. 2a displays the resulting 1D GIWAXS lineouts for each blend. Films with PCPDTBT lack any indication of lamellar polymer packing, whereas the fluorinated polymer contains a strong (100) peak at $Q = 5.5\,nm^{-1}$ indicating polymer aggregation[35]. Correspondingly, the distinct absorption of polymer aggregates at 800 nm[36] is very pronounced in the fluorinated blends but is lacking in the blends of the non-fluorinated polymer (see Supplementary Note 3), which is in general agreement with earlier findings[33]

In all blends, scattering at $Q = 15\,nm^{-1}$ suggests the presence of fullerene aggregation, as observed previously[19]. Coherence length for each fullerene peak was calculated with blends of

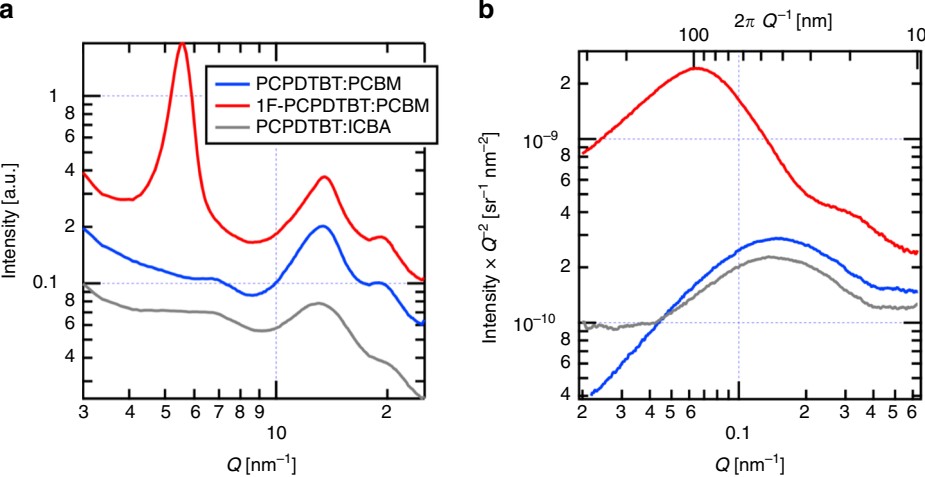

**Fig. 2** Scattering lineouts of the polymer:fullerene blends of interest: **a** GIWAXS vertical lineouts for all blends. Scattering intensity is shifted vertically to allow visual comparison of patterns. Peaks at $Q = 5 \text{ nm}^{-1}$ correspond to the polymer (100) plane, whereas the peak at $Q = 15 \text{ nm}^{-1}$ indicates the presence of fullerene aggregates. The final peaks at $Q = 20 \text{ nm}^{-1}$ are a convolution between the polymer (010) and a secondary fullerene peak, which precludes analysis of the polymer (010) packing in this case. **b** Lorentz corrected RSoXS profiles for all blends at 283.5 eV. Profiles are scaled to direct beam measurement to compare scattered intensity

PCPDTBT with PCBM and ICBA having coherence lengths of 1.44(2) and 1.09(2) nm respectively, indicating PCBM aggregates better than ICBA (see Supplementary Note 4 for the corresponding GIWAXS results of the neat components and the peak fits). Because of the short lengths compared with molecular size, these numbers should be considered the average distance over which ordering occurs rather than domain size. Blends with similarly broad fullerene peaks have been used to confirm the presence of fullerene aggregation within films[37]. Upon fluorination, the PCBM coherence length increases to 1.78(2) nm, suggesting even stronger fullerene aggregation promoted by increased polymer packing.

Mesoscale compositional domain structure was probed with resonant soft X-ray scattering (RSoXS). Lorentz corrected scattering lineouts taken near the carbon absorption edge are shown in Fig. 2b. Non-fluorinated PCPDTBT exhibits a single broad feature around $Q = 0.15 \text{ nm}^{-1}$ indicating subtle material fluctuations independent of fullerene choice[38,39]. By contrast, the fluorinated blend has an intense feature at $Q = 0.06 \text{ nm}^{-1}$ corresponding to a characteristic length scale of hundreds of nanometers. Additionally, a feature at $Q = 0.3 \text{ nm}^{-1}$ exists, likely due to a form factor resulting from polymer fibrils visible with atomic force microscopy (see Supplementary Note 5). Comparing the total scattering intensity from integrating the Lorentz corrected profiles gives a measure of the mean square fluctuations of the composition within the film[40]. From such analysis, the non-fluorinated blend has a mean square difference in domain composition that is at least four times lower than its fluorinated counterpart, suggesting a more intermixed morphology within the non-fluorinated blends.

**The role of excitation energy on free carrier generation**. PCPDTBT:PCBM blends were reported to exhibit a field-dependent charge generation yield[33] changing by 45% within a bias range from −2 to 1 V, making them the perfect candidate for ascertaining whether the field dependence changes as a function of temperature or excitation energy. Figure 3a displays the EQE spectrum of the blend. As reported previously, the EQE exhibits a distinct sub-bandgap feature, which we assign to charge generation via intermolecular CT states. The electroluminescence (EL) of that blend is also displayed, showing very little contribution

from the EL of the neat materials. Fitting the spectra with Gaussian line shapes yields an energy of the lowest energy excited CT state ($CT_1$) of 1.3 eV and a reorganization energy ($\lambda$) of 220 meV[41].

TDCF experiments were performed for selected excitation energies guided by the energy scheme in Fig. 3b. Importantly, illuminating the sample at 1.165 eV, near the EL emission maximum, will primarily excite low energy vibrations of the $CT_1$ state, whereas excitation at 1.29 eV (an excitation energy very close to $E_{CT}$) generates a vibronically excited CT state[18]. Similarly, an excitation energy of 1.55 eV was chosen to excite the lowest energy polymer singlet state $S_1$, while excitation with 2.33 and 2.95 eV generates higher energy excitons on the polymer and on the fullerene[8]. Special care was taken to ensure that only the selected wavelength is incident on the device (Supplementary Note 6). One might argue that an excitation energy of 1.165 eV (corresponding to 1064 nm) is still 60 meV above the CT emission maximum at 1.104 eV, meaning that we excite a state well above the vibronically relaxed $CT_1$. However, the shape of emission spectra is largely determined by ground state properties. According to the framework developed by Vandewal and coworkers[41,42], it is the steepness of the ground state potential around the nuclear coordinate of the fully relaxed $CT_1$ that governs the width of the CT emission while the emission intensity at a certain transition energy mirrors the Boltzmann population of the corresponding excited CT state. For PCPDTBT:PCBM, the EL intensity at 1.165 nm is only 15% lower than at the EL maximum, implying that illuminating the blend with this photon energy excites a CT-excited state within $k_BT$ of the fully relaxed CT exciton.

An example dataset of PCPDTBT:PCBM is shown in Fig. 3c, d, where TDCF data collected at different bias and temperatures are compared to the bias-dependent steady state current density under 1 sun equivalent illumination performed with a halogen lamp and adjusted for an intensity to yield the same generation current as at AM1.5 G illumination. At room temperature and above, the current–voltage characteristics nicely follow the field-dependent generation data, except for voltages near $V_{oc}$ where non-geminate recombination begins to play a role. The data compare well with previous TDCF measurements on a different PCPDTBT batch[33]. Lowering the temperature affects the current-voltage characteristics more severely than the TDCF

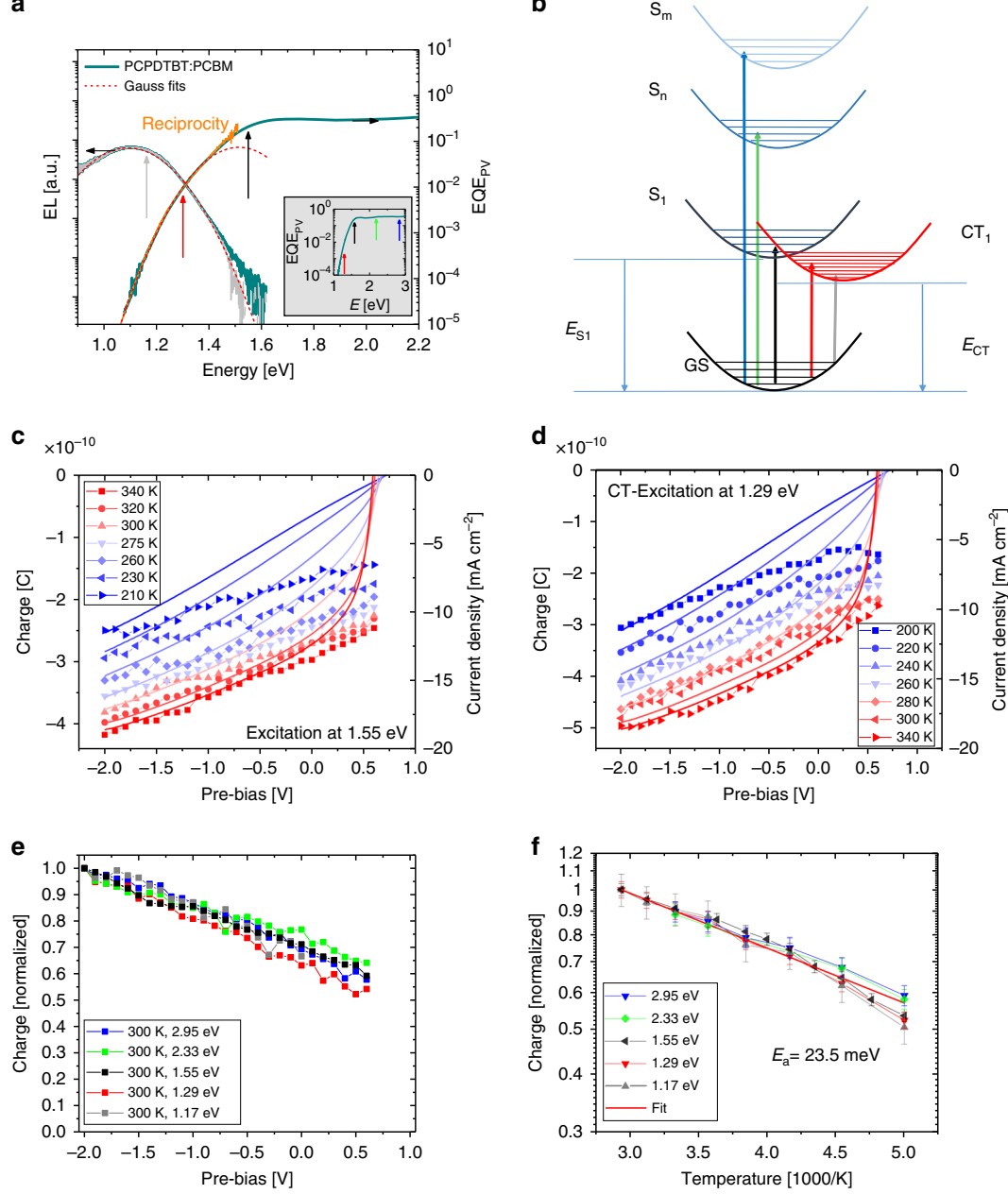

**Fig. 3** Charge generation as a function of field, temperature, and excitation energy: **a** EQE of the PCPDTBT:PCBM blend (dark green, right axis) compared to the electroluminescence spectrum measured for a low injection current of 0.1 mA cm$^{-2}$ (dark green, left axis), both at room temperature. Also shown is the EQE calculated from the EL spectrum with Rau's reciprocity relation (orange). The red lines are the Gaussian fits to the EL (minus the EL of the neat polymer) and low energy EQE, providing a CT energy of 1.3 eV. The arrows in the inset indicate the excitation energies in the TDCF experiments. **b** Schematic energy diagram of the PCPDTBT:PCBM blend with the optical transitions used in the TDCF experiments, arrows again indicate the excitation energies. **c**, **d** Bias-dependent charge generation (symbols) in a PCPDTBT:PCBM blend for different temperatures measured with TDCF for excitation above (1.55 eV) and below (1.29 eV) the optical band gap of the polymer. The collection voltage was −3 V, which is sufficient to completely extract photogenerated charges at all temperatures. Solid lines are current–voltage characteristics measured under illumination with a halogen lamp with the intensity adjusted to yield nearly the same current as under AM1.5 G illumination. **e** Photogenerated charge as function of bias at room temperature (normalized to its value at −2 V) for excitation at 2.95, 2.33, 1.55, 1.29, and 1.17 eV at room temperature. **f** Temperature dependence of the photogenerated charge in the low field plateau region for different excitation wavelengths. The solid line is the best fit to the data to an Arrhenius-type temperature dependence, yielding an activation energy of 23.5 ± 3.4 meV. Error bars indicate the standard deviation

generation data, particularly at low temperatures in the forward bias range, indicating increasing transport issues. At reverse bias, however, charge extraction becomes assisted by the internal electric field, and both sets of data merge. The field dependence of free charge generation plotted in a log-log scale exhibits the characteristic shape predicted and measured for D–A BHJ

blends[43–45], with an apparent plateau at lower fields followed by a continuous rise that becomes less steep with increasing temperature (Supplementary Note 7). Although applying analytical models to these data might be tempting, we note that none of these theories properly take into account the structural and positional disorder in such BHJ devices. Therefore, the following

analysis considers two parameters; the temperature dependence in the low field plateau region (where field-induced barrier lowering is insignificant) and the field dependence of generation at the lowest temperature measured. These are the cases in which hot generation pathways would be most easily detectable.

Figure 3e, f summarize these characteristic parameters for all employed excitation energies. Increasing the bias from $-2$ V to $V_{oc}$ decreases the efficiency of charge generation by nearly 50% (for additional bias-dependent generation data, see Supplementary Note 8). The temperature dependence follows an Arrhenius-type behavior with an activation energy of $23.5 \pm 3.4$ meV. Importantly, neither the field nor the temperature dependence are appreciably influenced by the excitation energy, even though this is varied from more than 0.3 eV below to 1.5 eV above the bandgap. These findings demonstrate that the majority of—if not all—free carrier generation occurs via thermalized states in this blend. A rigid proof of this interpretation is the direct excitation near the CT emission maximum, where only little extra energy is provided by the exciting photon. We note, however, a more pronounced reduction in the generation efficiency with decreasing temperature for small excitation energies in our TDCF results, albeit only at low temperatures. This might indicate weak contributions from non-thermalized (hot) CT states, but may also be caused by a certain narrowing (or shift) of the CT onset at low temperature. Importantly, no such effect is seen in the bias and temperature dependent data in Supplementary Note 8, pointing to a very small if any effect of excitation energy on the process of free charge formation.

Further support that the generation of free carriers comes from relaxed CT states is found in that the room temperature EQE in the sub-bandgap spectral range can be fully reproduced from the EL spectrum with Rau's reciprocity[46]. This suggests that the photogeneration of free charges (i.e., the measured EQE spectrum) involves the same thermalized states as their recombination (i.e., the measured EL spectrum)[18,19]. This conclusion contrasts with findings from earlier work regarding the role of excitation energy in the EQE of PCPDTBT:PCBM blends[47]. Here, EQE spectra measured at room temperature as function of bias revealed a stronger dependence of the photogeneration efficiency on bias below an excitation energy of *ca.* 1.5 eV. This study differs from ours in the way that the photogeneration efficiency was measured (steady-state EQE vs. TDCF), but also in the used polymer batch and in the final device efficiency (2 vs. 3% here), rendering a direct comparison difficult.

**Correlation of free charge generation with energetics and morphology.** As discussed above, replacing PCPDTBT with 1F-PCPDTBT or PCBM with ICBA both increases the energy of the charge separated state but affects the morphology in very different ways. Figure 4a, b compare the EQE and EL spectra of the 1F-PCPDTBT:PCBM and PCPDTBT:ICBA blends. Lowering the donor HOMO level (1F-PCPDTBT vs. PCPDTBT) or increasing the acceptor LUMO energy (ICBA vs. PCBM) increases the CT energy and consequently shifts both the EL emission maximum and the EQE onset to higher energies. We also notice the appearance of a weak shoulder at the high energy site of the EL emission of the blend of 1F-PCPDTBT with PCBM, which is assigned to the singlet exciton emission on well-aggregated polymer chains (see also Supplementary Note 9 for the

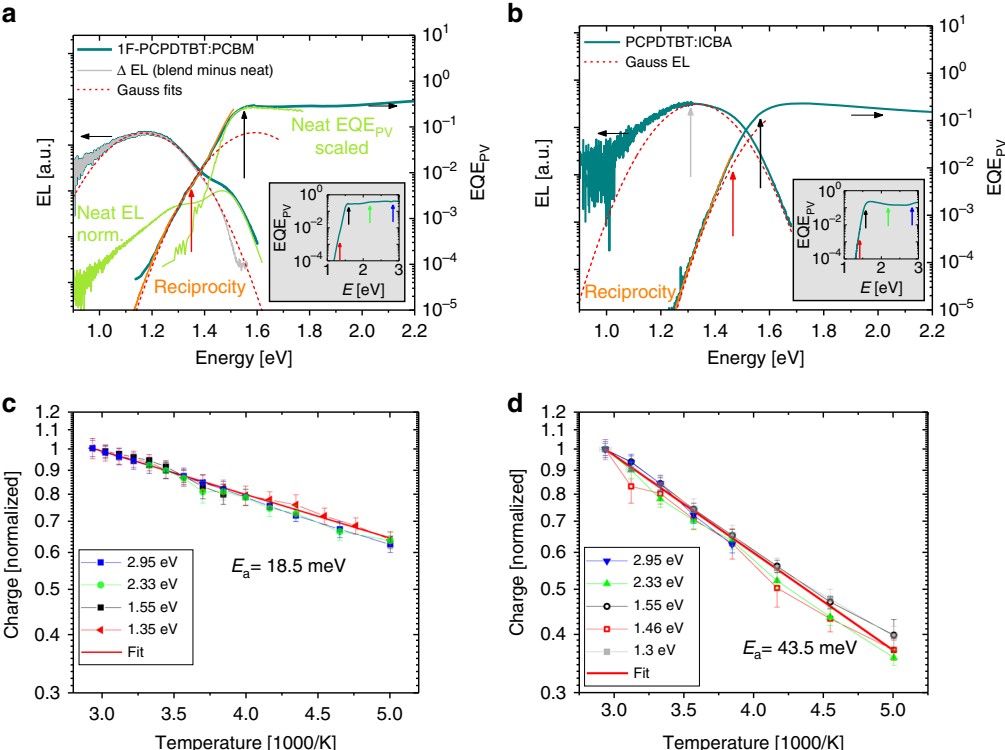

**Fig. 4** Charge generation as function of temperature and excitation energy: EL and EQE spectra of **a** the fluorinated 1F-PCPDTBT blended with PCBM and **b** the non-fluorinated PCPDTBT blended with ICBA. Also shown are the EQE and EL spectra of neat 1F-PCPDTBT, which are assigned to the polymer singlet exciton (light-green line). The gray lines show the blend EL corrected for the polymer emission. Red dashed lines show Gaussian fits to the low energy emission and EQE, revealing CT energies of 1.38 eV (1F-PCPDTBT:PCBM) and 1.52 eV (PCPDTBT:ICBA). Arrows in the gray inset indicate the excitation energies. Temperature dependence of the photogenerated charges in the low field plateau region for excitation below, at, and well above the polymer bandgap for **c** 1F-PCPDTBT:PCBM and **d** PCPDTBT:ICBA

discussion of the EL spectra[48]. By contrast, the EL of the blend of PCPDTBT with ICBA is broad, homogeneous, and independent of applied bias (Supplementary Note 9), indicating that it originates from CT emission only.

Temperature-dependent charge generation data are shown in Fig. 4c, d for below and above-bandgap excitation energies. The corresponding field-dependent data are discussed in Supplementary Note 8. Because of the blue-shifted EQE profiles, we were not able to get a reasonable set of TDCF data when exciting our sample at 1064 nm (1.17 eV), which is our strongest nanosecond laser source. In response to the reviewers, we have performed additional measurements with femtosecond excitation and, indeed, succeeded to retrieve a complete set of temperature and bias dependent TDCF data on the PCPDTBT:ICBA blend with excitation at 1.3 eV, which is exactly at the CT emission maximum (Fig. 4b). Importantly for our primary consideration regarding the role of excess energy in charge separation, we observe that the excitation wavelength has no appreciable effect on the field nor temperature dependence of free carrier generation in either of these systems (despite the very different morphology of the two blends). This is again supported by our observation that both blends follow Rau's reciprocity[46] throughout the entire sub-bandgap region, indicating that charge generation upon below bandgap CT excitation proceeds through the same state manifold as charge recombination leading to EL, and that the population of states can be described by Boltzmann statistics.

Both blends exhibit an Arrhenius-type temperature dependence but with the activation energy for the more disordered PCPDTBT:ICBA blend ($44 \pm 3$ meV) being significantly higher than for the much better ordered 1F-PCPDTBT:PCBM system

($19.5 \pm 2.0$ meV). This suggests the activation energy for CT separation to depend not just on the energetic structure of the heterojunction, but also on the mesoscale morphology (with disorder increasing the activation energy).

**CT recombination versus CT dissociation.** To conclude the findings from above, the insensitivity of the temperature and field dependence of free charge generation to the excitation energy suggests that the most free charge is generated though a common precursor, and that this precursor state is low in energy. As pointed out in the introduction, several papers suggested an energy-derived two-pool model, where photoexcitation either forms less-bound 'dissociable' or strongly bound 'localized' CT states, the latter one yielding most of the radiative geminate recombination. In view of this, we completed our study through an investigation of the emission properties of our PCPDTBT-based devices. The blend of PCPDTBT:ICBA is particularly well suited for this study because it exhibits the most pronounced effect of field (and temperature) on free charge generation, its CT emission is at short enough wavelengths to allow low-noise (accurate) measurements, and its morphology is highly intermixed. Previous studies suggested that such intermixed (disordered) blends exhibit the largest fraction of localized CT states[14–16]. Figure 5b shows steady state PL spectra of a PCPDTBT:ICBA device measured at different bias, compared to the EL spectrum of the same devices driven at 1.2 V. Importantly, the PL and EL spectra differ only little, with a slight blue shift of the PL. This slight blue shift of the PL might indicate non-complete thermalization of excitations in PL, which is

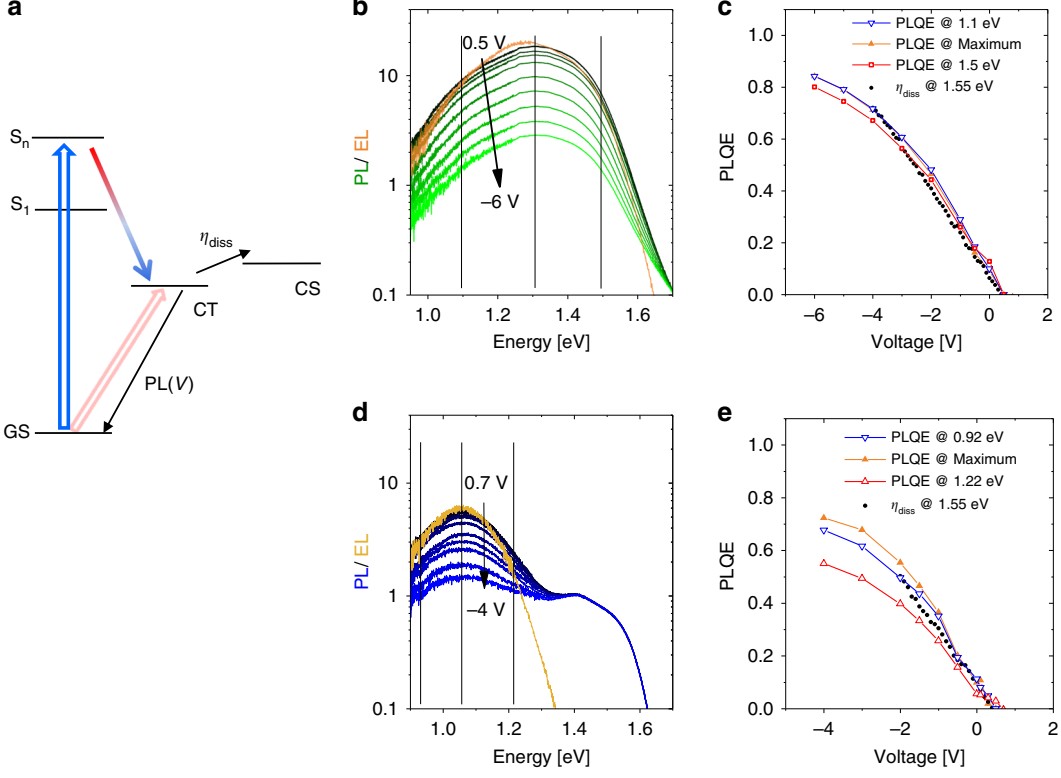

**Fig. 5** Bias-dependent photoluminescence; **a** energy scheme describing the competition between the recombination of the occupied CT state manifold, yielding PL with an intensity PL(V) and the dissociation of the very same manifold into free carrier with an efficiency $\eta_{diss}(V)$. **b** Bias-dependent PL data (green lines), compared to the EL driven at a bias of 1.2 V (orange line), **c** bias-dependent photoluminescence quenching efficiency PLQE(V), referenced to a voltage $V_0 = 0.5$ V for a 150 nm thick PCPDTBT:ICBA blend. PLQE data are plotted for three different detection energies, below, at and above the PL maximum (line-connected symbols). Solid black points display the prediction by Eq. 1 based on our bias-dependent TDCF experiments on the same devices. **d**, **e** the same for a 150 nm thick PCPDTBT:PCBM blend with the EL at 0.7 V

reasonable given the fact that photoluminescence occurs at shorter time scale than the recombination of free charge resulting in EL. Importantly, despite a pronounced effect of bias on the emission intensity, the shape of the PL is virtually unaffected by the applied voltage. These results suggest that only one state manifold governs PL, and that this is essentially the same that yields the electroluminescence. We now take into account our above finding that this blend obeys Rau's reciprocity, implying that electroluminescence proceeds via the same states as free charge formation.

Combined with the proven independence of the activation energy on the excitation energy, we conclude that photoluminescence and free carrier formation occurs via the same pool of low energy CT states, and that there is no separate pool of bound CT states (lying lower or higher in energy) which radiates PL but is essentially unconnected to the manifold of charge separated states. To support this conclusion, we made use of the fact that TDCF provides an accurate measurement of the field dependence of generation, down to low electric fields. If charges are generated by field-induced dissociation of the same pool of CT excitons that emits in PL (Fig. 5a), the quenching of the PL and the increase in the free carrier generation yield must be related through the following relation (see Ref. [49] and Supplementary Note 10):

$$\mathrm{PLQE}(V) = \left(\eta_{\mathrm{diss}}(V) - \eta_{\mathrm{diss}}(V_0)\right)/\left(1 - \eta_{\mathrm{diss}}(V_0)\right) \quad (1)$$

Here, $\mathrm{PLQE}(V) = 1 - \mathrm{PL}(V)/\mathrm{PL}(V_0)$ is the fluorescence quenching efficiency at a bias $V$ with regard to a reference voltage $V_0$ (with $\mathrm{PL}(V)$ being the bias dependent PL intensity) and $\eta_{\mathrm{diss}}(V)$ the CT dissociation efficiency.

Figure 5c shows $\mathrm{PLQE}(V)$ for three detection wavelengths, below, at and above the PL maximum. Values of the PLQE were calculated from the measured PL intensities with reference to a voltage $V_0 = 0.5\,\mathrm{V}$, which is safely below the EL onset of any of our devices. $\mathrm{PLQE}(V)$ increases almost linearly with decreasing bias, approaching values of above 80% in the considered bias range. To test the validity of Eq. 1, the bias dependent dissociation efficiency $\eta_{\mathrm{diss}}(V)$ was derived from a combination of bias-dependent TDCF data and photocurrent measurements with the assumption that the photocurrent saturation at large reverse bias corresponds to $\eta_{\mathrm{diss}}$ approaching unity efficiency (Supplementary Note 11). Figure 5b demonstrates that our bias-dependent PL and TDCF measurements indeed obey Eq. 1 providing firm support for our above conclusion that the state manifold involved in free charge generation also dominates the overall PL intensity. The situation is slightly more complicated for the more phase separated PCPDTBT:PCBM blend (Fig. 5d). While there is again a very good spectral overlap between the EL and the PL when both are measured near $V_{\mathrm{oc}}$, we also monitor a bias-independent PL at higher energies, probably stemming from polymer singlet emission. Accordingly, the course of $\mathrm{PLQE}(V)$ depends on the detection wavelength, with a weaker bias dependence at higher

photon energy (Fig. 5e). Considering only the PL quenching data at lower emission energies, Equation 1 is again well fulfilled. This result, combined with the agreement between the PL and EL spectra, the fulfillment of Rau's reciprocity and the insensitivity of the temperature and field dependent TDCF generation data to the excitation energy is inconsistent with the view that photoexcitation of this blend causes a significant occupation of low-lying, non-dissociable though emissive CT states under the given excitation and bias conditions. We also attempted to perform a similar study of the well phase-separated 1F-PCPDTBT:PCBM blend, as this particular system might exhibit pools of delocalized and localized CT states, well separated in energy and/or space. Unfortunately, the PL of this system was almost entirely dominated by the PL from aggregated polymer chains, and the PL intensity was only weakly affected by bias (Supplementary Note 12).

## Discussion

Our data reveal that the excitation energy has a small to negligible effect on how field and temperature impact free charge generation: direct excitation of the low energy CT states leads to virtually the same field and temperature dependence as excitation far above the band gap. From this we conclude that free charge generation does not benefit from the excess energy provided to the system via excitation into higher lying states. This conclusion is valid irrespective of variation in the CT energies when PCPDTBT is replaced with 1F-PCPDTBT or PCBM with ICBA, which moves the CT state closer to the energy of the relaxed singlet exciton on the polymer.

We conclude that charge generation proceeds through a common state located at an energy below that of the polymer singlet state, the thermalized CT manifold. Also, our PL and EL data suggest that this very same state manifold yields most of the CT photoemission, meaning that the dissociation of such low energy CT states competes directly with its geminate recombination. As such, the efficiency of CT dissociation, and the mechanism by which electric field and temperature boosts this process depends on the details of the energetic and mesoscale morphological landscape at the heterojunction, e.g., the shape and heights of the energy barrier towards free charge formation, and its spatial and energetic disorder. This is why the choice of the components (and the preparation conditions) has such a strong effect on the charge generation efficiency[50]. Our data also support the depiction that the presence of pure extended domains in the donor or acceptor favors free charge formation[13,16,17]. Indeed, Table 1 indicates a systematic anti-correlation between the activation energy and the fullerene correlation length. On the other hand, there is no unambiguous relation between the activation energy and the driving force for electron transfer, $\Delta E_{\mathrm{ET}}$, which is the difference between $E_{\mathrm{S1}}$ and $E_{\mathrm{CT}}$ (Table 1). This indicates that the excess energy provided to the system via a higher $\Delta E_{\mathrm{ET}}$ alone is not necessarily sufficient for reducing the effective barrier for charge formation in a blend system.

**Table 1 Parameters describing the temperature and field dependence of charge generation relative to energetics and morphology from left to right**

| Blend | Activation energy (meV) | Field dependence (%) | Driving force (meV) | Fullerene correlation length (nm) |
|---|---|---|---|---|
| 1F-PCPDTBT:PCBM | 19.5 ± 2.0 | 27 ± 5 | 125 ± 10 | 1.78 ± 0.02 |
| PCPDTBT:PCBM | 23.5 ± 3.4 | 49 ± 6 | 209 ± 10 | 1.44 ± 0.02 |
| PCPDTBT:ICBA | 44 ± 3 | 65 ± 5 | 70 ± 10 | 1.09 ± 0.02 |

Activation energies of free charge formation taken from Figs. 3f and 4c, d
Bias dependence of charge generation at 200 K and below bandgap excitation, expressed as the percentage decrease of collected charge when increasing the pre-bias from −2 to 0.4 V
Driving force for electron transfer, which is the energy difference between the relaxed singlet exciton on the donor and of the CT state
Correlation length of the fullerene clusters from GIWAXS

In turn, our data show that neither a large excess energy provided by the excitation of high energy singlet states nor an appreciable offset between the energy levels of donor and acceptor provides conditions to bypass low-lying CT states, irrespective of whether the blend is highly intermixed (as in the PCPDTBT:ICBA blend) or phase-separated (as in the 1F-PCPDTBT:PCBM blend). With that, our results put strong emphasis on the energetics and dynamics of these low energy interfacial excitations. These properties were shown to depend, inter alia, on the details of the molecular conformation and arrangement of the donor and acceptor molecules at the DA heterojunction[51,52]. Through this mechanism small changes of the molecular structure can make a large impact on the efficiency of charge generation. A recent example of this is the large effect on the temperature and field dependence of free charge formation of donor molecule alignment at a structurally well-defined planar heterojunction with $C_{60}$[53]. Quantum chemical simulations revealed that this dependence arises from distinct differences in the electronic coupling between the lowest energy charge transfer states and the ground state of the two different orientations. Such planar heterojunctions may be regarded as model systems for the bulk-distributed heterojunction in efficient phase-separated blends.

We also find that activation energies for free charge formation are comparable to or even below the room temperature thermal energy. This observation is consistent with recent results from temperature dependent EQE or PL quenching experiments on various polymer:fullerene blends[21,29]. With these activation energies being independent of the excitation energy, they are a measure of the extra thermal energy which is needed to dissociate the thermalized CT state population into free charge carriers, a process that is in direct competition with geminate recombination of exactly the same population. A direct correlation between the activation energy and the binding energy of the contributing states is, unfortunately, not trivial. While for a homogeneous 3D system Braun's model predicts the zero field dissociation rate to follow an Arrhenius-type temperature dependence, with the activation energy being the CT binding energy, the situation becomes more complicated in presence of energetic disorder. Here, additional low energy states become available for free charges, resulting in a concurrent reduction of the apparent activation energy[22,23,54]. However, our observation of the mostly ordered system having the lowest activation energy questions a predominant effect of disorder on the activation energy. Recent transient electroabsorption measurements revealed efficient CT splitting in a polymer:fullerene blend with very little energetic disorder[55]. We propose that our activation energies, decreasing from around 44 meV for the highly intermixed PCPDTBT:ICBA blend down to below 20 meV for the phase-separated 1F-PCPDTBT:PCBM system mirror a significant reduction in the CT binding energy due to charge delocalization at the heterojunction[44,56,57], microelectrostatic effects[58,59], or the stabilization of free charge on the pure donor and/or acceptor[60–62]. Notably, recent mesoscale quantum mechanical modeling predicted the lowest interfacial CT state energy to be of the order of thermal energy[63]. This work also highlighted the importance of both carriers being able to delocalize for a weak binding of the lowest energy CT state. In fact, some of the most efficient organic solar cells display phase-separated morphologies with a high molecular ordering in both the donor and the acceptor rich phase[1,3,64]. Our results suggest that the superior performance of such blends with well-aggregated donor and acceptor phases arises from a weak binding of low energy CT states, rather than from free charge generation through hot dissociation pathways. Clearly, further work is required to generalize this statement for a wide material basis.

In conclusion, we find that neither the temperature nor the field dependence of generation is affected by the excitation energy, despite the fact that the photon energy was varied from direct CT excitation to nearly 1.5 eV above the polymer optical bandgap. Notably, activation energies deduced from our temperature dependent measurements lie in the range of few tens of meV, with the lowest value for the most phase-separated blend. We also find that the electroluminescence, photoluminescence, and charge generation properties are at variance to a two pool model where geminate recombination losses proceed mostly through a separate pool of strongly bound (non-dissociable) CT states. We conclude that photoexcitation in our blends populates a low energy energy (thermalized) CT manifold, independent of photon energy, that this manifold is the precursor of most free charge and CT photon emission, and that fairly little thermal energy is needed to render CT dissociation competitive to geminate recombination. With that, this work provides an important contribution to the long-running debate regarding the predominant pathway of generation in organic solar cell devices, and also clear guidance for future simulation work regarding the mechanism of free charge formation.

## Methods

**Sample preparation.** PCPDTBT was purchased from 1-Material and has a molecular weight (Mw) of ca. 48 kDa with a polydispersity (PDI) of 1.31. The alternating copolymer [2,6-(4,4-bis(2-ethylhexyl)−4H-cyclopenta[2,1-b;3,4-b′] dithiophene)-alt-4,7-(5-fluoro-2,1,3-benzothiadiazole) (1F-PCPDTBT) was synthesized by microwave-assisted Stille cross-coupling polymerization at Fraunhofer IAP as it was described in the ref.[33]. The number-average molecular weight ($M_n$) is around 38 kDa with a polydispersity of 1.46 as determined by high-temperature gel-permeation chromatography (GPC) at 135 °C in trichlorobenzene at the MPI for Polymer Research in Mainz. The solvents chlorobenzene and the processing additive diiodooctane (DIO) were bought from Sigma Aldrich. PCBM and ICBA were purchased from Solenne BV and have a Mw of 1031 and 953 Da, respectively.

The devices were prepared on structured ITO (Lumtec) substrates coated with 30 nm PEDOT/PSS (Clevios AI 4083). The blend solution (1/3 by weight) of PCPDTBT or 1F-PCPDTBT and PCBM or ICBA in chlorobenzene was spincoated yielding an active layer thickness of 110 nm for temperature dependent TDCF and 150 nm for PL quenching measurements. The samples were finalized by thermal evaporation of Ca (20 nm) and Al (100 nm) with an active area of 1.1 mm² for TDCF experiments and 16 mm² for EQE, EL and JV measurements. All samples were encapsulated with epoxy resin and a glass lid prior to air exposure.

**Grazing-incidence wide-angle X-ray scattering and resonant soft X-ray scattering.** GIWAXS was conducted at beamline 7.3.3 at the Advanced Light Source (ALS), Lawrence Berkeley National Laboratory using 10 keV photons at an incident angle of 0.2°. Samples were cast onto silicon substrates coated with PSS to replicate device morphology. Details on instrumentation can be found elsewhere[65]. RSoXS measurements were carried out at beamline 11.0.1.2 at the ALS[66]. X-ray energy was chosen to maximize phase contrast between components and limit background signal from X-ray fluorescence. Samples were floated from silicon substrates coated in water-soluble PSS in deionized water onto silicon nitride windows for RSoXS measurements.

**Temperature dependent time delayed collection field experiments.** In TDCF, we excite the device with a suitable wavelength at a varying pre-bias, followed by collection field. A fs-TiSa Laser (Coherent Libra 4 mJ cm⁻², 800 nm) pumps a Coherent Opera Solo NOPA, which provides various excitation energies. Special care was taken to ensure that only the wavelength needed hits the sample and that there is no additional non-linear effect (e.g., second harmonic generation (SHG)) in the device. Therefore, the spectral response was measured with a highly sensitive partially home-build UV-NIR spectrometer (electronics by Entwicklungsbüro Stresing, concept, optics and design by Jona Kurpiers and Florian Dornack). For excitation of the CT-maximum a ns-Nd-YAG Laser (Ekspla) was used to successfully eliminate SHG (only CT-Photoluminescence was observed). The pre-bias and collection bias was provided by a fast, low jitter function generator (Agilent 81150 A) which is directly triggered by a delay generator (Coherent SDG). Special care was taken to avoid free carrier recombination prior to extraction. Therefore, mild excitation conditions and also fast ramp-up of the bias is applied. Fast charge extraction needs a small measurement resistor (10 Ohm) and a small capacitance (~100 pF) which results in a short and large recharging current of ca. 1 Ampere within 2.7 ns. A home-build measurement amplifier was used to apply the pre-bias and collection bias and measure the current through the device[32]. As we aimed at the study of charge generation for a wide range of excitation energies, possibly including direct excitation of the vibronically relaxed CT state, similar carrier densities were ensured despite the fact that the EQE varied by 5 orders of

magnitude, down to $10^{-4}$, in the photon energy range studied here. Therefore a high definition oscilloscope (Agilent DSO9104H) with very low noise input amplifiers in combination with a home-written software lock-In amplifier was used to reach the high sensitivity. Computer custom codes are available from the corresponding author upon reasonable request. The device was positioned on the cooling finger of a closed cycle helium cryostat (ARS-CS202-X1.AL). Electrical connection was realized through a home build, directly attached amplifier as close as possible to the device, with the drawback of loosing cooling power and reaching only 200 K. The cryostat was heated and evacuated to $1.3 \times 10^{-5}$ mbar (Pfeiffer TCP121 Turbo pump and Edwards XDS-10 scroll pump) over night to remove residual water from the vacuum chamber prior the temperature dependent measurements.

**External quantum efficiency and absorbance.** Broad white light from a 300 W Halogen lamp (Phillips) is chopped (thorlabs MC2000), guided through a cornerstone Monochromator and coupled into an optical quartz fiber, calibrated with Newport Photodiodes (818-UV and 818-IR) and a SR 830 Lock-In Amplifier which measures also the response of the solar cell.

Absorbance was measured with a Varian Cary 5000 in double beam mode in transmission and with an integrating sphere in reflection.

**Electro- and Photoluminescence.** For electroluminescence measurements the device is hold at a constant voltage (Keithley 2400) for 1 s. The emission spectrum was recorded with an Andor Solis SR393i-B spectrograph with a silicon detector DU420A-BR-DD and an Indium Gallium Arsenide DU491A-1.7 detector. A calibrated Oriel 63355 lamp was used to correct the spectral response. EL spectra where recorded with different gratings with center wavelengths of 800, 1100, and 1400 nm and merged afterwards. The same setup was used for bias dependent photoluminescence measurements, under steady state illumination with a 635 nm laser, corresponding to short a circuit current of 0.15 mA cm$^{-2}$. Note that the bias dependent PL experiments were performed on thicker (ca. 150 nm) blends, as those allowed almost complete quenching of the PL without the appearance of strong leakage currents. Because of microcavity effects, the PL and EL spectra of these 150 nm thick samples are slightly red-shifted compared to the EL spectra of the 110 nm thick samples, shown in Figures 2 and 3.

**Temperature dependent time resolved photoluminescence.** The temperature of the sample was controlled by a closed cycle cryostat (Oxford Instruments, Optistat Dry TLEX). For excitation a mode-locked Ti:sapphire laser (Coherent, Chameleon Ultra) with a pulse width of 140 fs and a repetition rate of 80 MHz was used. The photoluminescence transients were recorded by a Hamamatsu Universal Streak Camera C10910 coupled to an Acton SpectraPro SP2300 spectrometer. The camera was used in synchroscan mode to allow for a 2 ns time window and a FWHM of the instrumental response function of 44 ps.

**Data availability.** The datasets generated and analyzed during the current study are available from the corresponding author on reasonable request.

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

## Acknowledgements

We thank Burkhard Stiller for AFM pictures, Andreas Pucher for support with electronics, Eileen Katholing for support in synthesizing 1F-PCPDTBT and Dominik Gehrig for GPC analysis. D.N. and J.K. acknowledge Koen Vandewal and Andreas Hofacker for fruitful discussions. This work was funded by the Deutsche Forschungsgesellschaft (DFG) Projekt Nr. NE 410/13-1, NE410/15–1, INST 336/94–1 FUGG.

## Author contributions

J.K. and S.R. prepared the samples and devices used for all measurements. J.K. performed all TDCF measurements, S.R., T.T., S.A., and J.K. performed P.L., E.L., E.Q.E. and absorbance. S.J. synthesized 1F-PCPDTBT. T.F. and B.A.C. completed all X-ray diffraction measurements and analysis for structural characterization. I.H. and M.J. performed temperature dependent transient PL measurements. F.J. was involved in the design and advancement of the used laser equipment. D.N. supervised the project. D.N., J.K., T.F., and S.R. wrote the manuscript, and all authors revised the final manuscript.

## Additional information

**Competing interests:** The authors declare no competing interests.

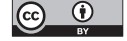

