## [Peer Review File · Nature Communications]

Reviewers' Comments:

Reviewer #1:

Remarks to the Author:

The manuscript by Kurpiers et al presents a deep and comprehensive photophysical study of charge photogeneration in the blends of PCPDTBT derivatives and fullerenes. Authors used a combination of methods including quite unique delayed extraction technique to evaluate the binding energies of states recombining within the first 4 ns after charge photogeneration. They come to the conclusion that binding energies do not depend on excitation wavelength. Authors interpret this as dominantly cold separation of so-called interfacial CT states.

Although I have some concerns with methodology (see below), I find it a very creative and solid piece of research which definitely deserves good visibility in the community. However, to apply the techniques of choice, authors select materials with low efficiency (3%) and poor morphology (except one system). This makes it unclear how well the results can be generalised on other, more efficient, OPV systems.

I would recommend the manuscript for a good specialized journal, even the level of JACS. But, for above-mentioned reason, I am not fully sure it fits Nature Com.

Specific comments:

1) the study is made mostly on the devices with very strong field dependence. This is probably an evidence of poor morphology and it is not clear what kind of recombination is happening in the first 4ns. Are those interfacial states recombining? or 'gold fish' states when charges are locally free but trapped in the domains of very small size.

Intuitively, I would not expect substantial 'hot' generation in such materials anyway as 'hot' process takes advantage of charge delocalisation into polymer or fullerene aggregates. I believe authors should be more specific about on what type of systems their conclusions can be generalised...

2) in connection to (1) - very nice control experiment is shown in figure 5. But it is performed on the material with the worst morphology... can this be also done on the crystalline system, where hot process is more likely?

3) I could not find any details or control experiments on the strong extracting bias. How strong is it and how sensitive is the experiment to the level of bias used? Also the effect of delay time before extraction on the experimental results can give additional insight into which binding energy is actually measured.

Reviewer #2:

Remarks to the Author:

In the field of organic solar cells there is already for some years an ongoing debate regarding the route of how exciton dissociation occurs : through so-called "hot" CT-states or through thermally relaxed "cold" CT-states. In this manuscript the authors address this challenging issue through the study of the so-called "activation energy for free charge photogeneration" using mainly time-delayed-collection-field (TDCF) experiments. The Neher group is recognized for its expertise in this domain and they have used this technique already in the past to support the view that charge generation in organic solar cells occurs through the relaxed CT states (cfr. reference 26). Of particular interest in this manuscript is the performance of temperature and field dependent experiments yielding the above mentioned "activation energy for free charge photogeneration" which results to be of the order of thermal energy and nearly independent of the excitation energy. Furthermore, the authors highlight the importance of morphology comparing

donor:acceptor blends using fluorinated and non-fluorinated PCPDTBT as donor material and PCBM and ICBA as acceptor material yielding LUMO and HOMO offsets in the same direction but with different effects on morphology.

Although this work is in general solid and of interest for the OPV-community, the following concerns and remarks are formulated :

The main concern regarding this manuscript is that the 'meaning' of the obtained "activation energy" is not sufficiently clarified. In the manuscript a perhaps disproportionate part is devoted to the description of the morphology, while the core issue of the nature, meaning and effects of the obtained "activation energy" could be discussed more extensively. On page 6 the authors state that the activation energy "is a reasonable measure for the binding of the lowest state involved in the pathway of charge generation". On page 21 the authors write "given that our activation energies are independent of excitation energies, such values must represent the effective binding of low lying "thermalized" CT state populations. The formulations regarding the nature of the "activation energy" seem to be insecure and a more extensive description could improve the formulated concern.

On page 17 (from line 23) the different activation energies are mentioned but no discussion is provided on why these activation energies are different.

The energy of the relaxed CT state is also temperature dependent (cfr. Vandewal et al., Physical Review B, 2010). This effect is seemingly not considered in the present work. How does this affect the proposed results ?

On page 19 – line 15 – the intrinsic disorder in N2200 is mentioned (33 meV). Why is the energetic disorder not listed in Table 1 – p. 20 ? What is the relation between the energetic disorder and the obtained activation energies ?

In relation to the discussed temperature dependence of electron transfer in PCPDTBT:PCBM, we would like to signal to the authors the recent paper by Unger et al. on "The impact of driving force and temperature on the electron transfer in donor-acceptor blend systems" in Journal of Physical Chemistry C – October 2107.

The investigated donor PCPDTBT is not among the current high performance materials. Furthermore, in table SI1 the authors give a list of material systems and their specific issues in relation to the presented TDCF measurement methodology. How general are therefore the obtained results ?

Why did the authors not insert the IQE curves for the investigated blends ?

Minor remarks :

Reference 27 and reference 61 are the same.

Consequent format of references.

Please check carefully the text again – see for instance :

Page 16 – line 2 : optical bad (= >band) gap

Supporting information p. 1 – line 21 : Table SI1: Overview of other blends with TDCF and the reason why these blends were NOT (NOT is missing) considered further.

Supporting information p. 3 – line 12 : the absorption of the neat films is reveals ??

Supporting information p. 3 – line 18 : the affect (=> effect) of fluorination

Supporting information p. 3 – line 18 : more pronounced then (=> than)

Supporting information p. 6 – line 9 : will the surface => while the surface

Supporting information p. 8 – line 2 : change generation => charge generation

Y-axis in 3 figures of figure S18 : => same division

...

As mentioned before, this work is in general solid and of interest for the OPV-community, but based on the formulated concerns and questions, we are not convinced that the presented manuscript will 'resolve this debate' ('hot' vs 'cold' CT states) as aimed by the authors in the abstract. Furthermore, we do not consider the presented results and methodology sufficiently groundbreaking (cfr. references 36,40) for publication in Nature Communications, but would recommend publication in a more specialized journal.

Reviewer #3:

Remarks to the Author:

The paper by Kurpiers et al reports on the field dependence of charge photogeneration in polymer-fullerene solar cells, i.e. the field dependent dissociation of (slightly) bound electron-hole pairs at the donor-acceptor interface to extractable charge carriers. The authors state that they finally settle the dispute on the importance of excess energy (hot generation) on the charge photogeneration from relaxed electron-hole pairs. While I appreciate the experimental evidence provided by the authors and its added value (see below), I believe this claim to be somewhat too strong. In 2010, Lee et al showed this already in a JACS paper, and it was confirmed later by e.g. Vandewal et al in Nature Mater 2014 with contributions from the Neher group. The reported measurements in these publications were close to solar cell operation conditions, in contrast to the measurements by Grancini et al and Bakulin et al which both used fs-spectroscopy. However, the data shown by Bakulin et al did not, I think, prove that hot generation was dominant, as they even reported its comparatively small relative contribution and highlighted the importance of delocalisation (while saying that higher energy states might be more delocalised). While I would ask the authors of the present paper to state that the verified prior work with conditions close to solar cell operation, instead of claiming to settle a dispute, I am still in favor of publishing the reported research in Nature Communications: The findings on the small (thermal) activation energy by a highly relevant and sensitive technique are very important and relevant to researchers in the field. They can be a reliable basis to describe, e.g. by theoretical work, what finally is the dominant driving force for dissociation of bound pairs (or why the binding energy is so low). Also, the additional study on the potential difference of electroluminescence and photoluminescence in view of the model of two different CT state types is very interesting.

From an experimental perspective, the work is also very relevant to working solar cells due to the low laser fluences used in this study. However, I am a bit concerned about the used laser excitation with different photon energies.

First, let me point out that this is not the first time this was done, the authors and others also published photon energy dependent data of charge extraction on organic solar cells. I stress that here as the authors state "Surprisingly, we find a weak to negligible effect of the excitation energy on the charge generation activation energy.". As a very weak impact of photon energy on field dependent photogeneration was already reported in literature, replicating this finding (even when

using another donor material) should have been not very surprising.

Second, the lowest photon energies used in the study, while in the range or even below the CT state energy defined as half way between emission and absorption peak, are still above the relaxed CT energy as given by the electroluminescence emission peak. For instance, the authors use photon energies of laser excitation of 1.17 eV or more, where the relaxed charge carriers reside around 1.1 eV as measured by EL. These are 3 kT difference even at room temperature for the PCPDTBT. It is much more for the fluorinated version which was excited at 1.35 eV, even if the 1.17 eV laser photon energy used before would have been a perfect fit to the relaxed CT energy of (I am estimating) 1.17 eV. This implies that the directly excited bound pairs have an excess energy of several kT and are not perfectly relaxed. This shortcoming is shared with all previous studies applying pulsed lasers, by the way.

A note concerning the authors claim that transient absorption cannot distinguish (always) between charges and triplets. That might be so in special cases, but usually they can be separated, as shown for instance by Chow et al, JACS 2014, or the work by Laquai and Howard. Also, the authors state that the sensitivity of TDCF exceeds that of transient absorption. This might also be the case, but I want to point out that transient absorption data over many orders of magnitude in time and charge have been presented, whereas the data in the current manuscript is only shown on a linear scale in terms of charge. I assume the authors refer to the low excitation fluence of their technique, which is indeed a special trademark of the technique and the excellent experimental optimisation of the authors. That said, looking at the charge on a linear scale, I am not sure that dispersive non-geminate recombination can be ruled out, although its importance for the solar cell working conditions likely will be very limited for the shown material system. Also, it is not the focus of this manuscript.

In Fig. 3 f the authors show the field dependence of charge generation. It is unclear to me which voltage bias was used for this cross section of the data, as I cannot identify the "low field plateau region" mentioned in the text.

Finally, let me point out again that while I raise some concerns which should be commented upon by the authors, I believe that this study is a very important contribution on charge photogeneration, in particular in view of the very small thermal activation energies; and this for a material system showing a rather pronounced field dependence in contrast to systems such as P3HT or PTB7 blended with fullerene. Also, the careful consideration of the difference between electroluminescence and photoluminescence, reported almost ten years ago, has been verified with high sensitivity under low injection currents to ascertain comparable conditions. In my opinion, this is an important contribution to the discussion around the model with the two CT reservoirs.

Carsten Deibel

Replies to the Reviewers

General Comments

Thank you very much for your helpful comments. We have now revised the manuscript, taking into account the issues raised in your report. Our detailed replies are listed below. Major changes to the paper in reply to the Reviewers' reports are highlighted in yellow in the main manuscript.

In general, the three Reviewers were concerned about the novelty of the work and the generality of our conclusions. In Reply to this, we have reordered and rewritten part of the manuscript, putting more emphasis on the aims of our study, but also added results from additional PL experiments. On the other hand, we have shortened the chapter on the structural characterization and removed the generation data of the neat N2200 film. We, further, altered the title of the paper, now putting more focus on the pathway of free charge generation which is the main objective of this paper.

Regarding the novelty of our work, we acknowledge that we and others have previously performed TDCF studies on different polymer:fullerene blends with varying excitation energy. However, all of these previous experiments were performed within a limited energy range (mainly because of the limited sensitivity of our "older" TDCF setup), and all of our experiments were performed at room temperature. With our new step, direct CT excitation (around the EL maximum) has now become possible, but also allowed us to perform accurate measurements of the temperature dependence of generation as a sensitive tool of investigate the energetic pathway of generation. We find that for all three blends studied here, this temperature dependence is not affected by excitation energy, though we varied this energy from direct CT excitation to almost 1.5 eV above the optical bandgap. We completed this study by a comparison of EL and PL properties, and analyzed the pronounced bias dependence of PL with regard to the efficiency of free charge generation. In combining these results, we come to the important conclusion that photoexcitation in our blends populates a low energy (thermalized) CT manifold, independent of photon energy, that this manifold is the precursor of most free charge and CT photon emission, and that fairly little thermal energy is needed render CT dissociation competitive to geminate recombination.

Regarding the suitability of the studied system and the generality of our conclusions we point out that the manuscript is submitted as a Communication, where we apply a dedicated method to a well-selected material system to address an important scientific question. This is not a full article published in one of the main Nature Journals and it is not intended (and suited) to provide universal answers. Taken this into account, we altered some the final statement regarding the generality of our findings. In particular, the statement in the original version that we settled the dispute regarding the contribution by hot generation pathways might have been a bit too strong, given the fact that we did not study a wide range of materials.

Detailed comments

Reviewer #1 (Remarks to the Author):

The manuscript by Kurpiers et al presents a deep and comprehensive photophysical study of charge photogeneration in the blends of PCPDTBT derivatives and fullerenes. Authors used a combination of methods including quite unique delayed extraction technique to evaluate the binding energies of states recombining within the first 4 ns after charge photogeneration.

They come to the conclusion that binding energies do not depend on excitation wavelength. Authors interpret this as dominantly cold separation of so-called interfacial CT states.

Although I have some concerns with methodology (see below), I find it a very creative and solid piece of research which definitely deserves good visibility in the community.

However, to apply the techniques of choice, authors select materials with low efficiency (3%) and poor morphology (except one system). This makes it unclear how well the results can be generalised on other, more efficient, OPV systems.

I would recommend the manuscript for a good specialized journal, even the level of JACS. But, for above-mentioned reason, I am not fully sure it fits Nature Com.

First of all, we thank the reviewer for the overall positive feedback. We are well aware of the fact that today's polymer-based blends reach efficiency levels well above what is reported here. However, our choice of the material system was guided by some specific requirements, some which are already outlined in the manuscript.

- a. The system should have a broad absorption range, allowing us to vary the excitation energy over a wide range, from the CT state to highly excited excitons.*
- b. The system should exhibit an appreciable field dependence of generation, where we also expect a significant effect of temperature on the generation efficiency.*
- c. The material platform should be chosen as such that the morphology and performance can be easily controlled by small changes of the chemical structure*
- d. The system should be free of dispersive effects, as this would aggravate the determination of the free carrier generation efficiency from the TDCF experiments.*
- e. The fullerene (acceptor) cluster size should be small enough to not give any own contribution to the field dependence of generation.*

*Fortunately, all these conditions were safely met by our PCPDTBT:fullerene platform. Notably, our 1F-PCPDTBT:PCBM system exhibits a fairly decent efficiency (mostly limited by the low Voc). Regarding the overall interest in this polymer system, we note that PCPDTBT was among the first low-bandgap copolymers with decent efficiencies, and that there is an ongoing interest in understanding and employing the photophysical and photovoltaic properties of this polymer and related compounds. We also like to point out that this manuscript has been submitted as a Communication, where we apply a dedicated method to a well-selected material system to address an important scientific question. This is not a full article published in one of the main Nature Journals. As a last comment, we emphasize that several papers (full articles and communications) appeared recently in Nature journals, which employed organic systems with similar or even poorer performance as reported here to address a specific question of high interest, and that those articles have attracted considerable interest in the community. (e.g. M. Causa et al., Nat. Commun. **7**, 12556 (2016); J. Benduhn et al., Nat. Energy **2**, 17053 (2017); N.A. Ran et al., Nat. Commun. **8**, 79 (2017). In response to the comments by the Reviewer, we have, however revised part of the manuscript describing why this material has been chosen. We also altered some the final statement regarding the generality of our findings. In*

particular, the statement in the original version that we settled the dispute regarding the contribution by hot generation pathways might have been a bit too strong, given the fact that we did not study a wide range of materials. But once again, this is a Communication, it's not intended (and suited) to provide a universal answer.

Specific comments:

1) the study is made mostly on the devices with very strong field dependence. This is probably an evidence of poor morphology and it is not clear what kind of recombination is happening in the first 4ns. Are those interfacial states recombining? or 'gold fish' states when charges are locally free but trapped in the domains of very small size.

Intuitively, I would not expect substantial 'hot' generation in such materials anyway as 'hot' process takes advantage of charge delocalisation into polymer or fullerene aggregates. I believe authors should be more specific about on what type of systems their conclusions can be generalised...

Thank you very much for these important comments. It is, indeed, an intriguing observation that in both morphologies, the highly intermixed and the extremely aggregated blend the charge generation is not depending on the photon energy. The different morphologies are made clear in the main text and in figure 2; there is no polymer aggregation peak for blends with PCPDTBT, but a distinct peak at $q \approx 5 \text{ nm}^{-1}$ corresponding to the strong aggregation of 1F-PCPDTBT. Also AFM pictures in figure SI5 show a needle like morphology for the blend with the fluorinated polymer. To answer your question about what is happening in the first 4 ns, we have extended the discussion and documentation of the TDCF recombination measurements. We also entered a collaboration with Ian Howard and Marius Jakoby to measure time resolved photoluminescence of the CT-exciton on our blends. The data show that there are basically no long-lived dissociable CT states which survive the first 4 ns (which is when extraction starts), meaning that TDCF with a 4 ns delay actually measures only the free (extractable) charge which is formed in competition to geminate recombination. We can, however, not rule out that some photogenerated charges become trapped on isolated domains, but we also highlight the very good correspondence between the bias dependent generation data and JV characteristics at reverse bias.

In response to this question, we have extended Figure SI1 (also now including transient PL measurements) and extended the main text accordingly.

2) in connection to (1) - very nice control experiment is shown in figure 5. But it is performed on the material with the worst morphology... can this be also done on the crystalline system, where hot process is more likely?

In response to your comment, we attempted to do the same analysis with the well phase-separated and more crystalline 1F-PCPDTBT:PCBM blend, as this particular system might indeed exhibit pools of delocalized and localized CT states, well separated in energy and/or space. Unfortunately, the PL of

this system was almost entirely dominated by the PL from aggregated polymer chains, and the PL intensity was only weakly affected by bias. The latter is expected because this is the material with the most efficient generation at low fields. Because of these difficulties, we abstained from a more detailed analysis of the emission data. A further complication was that we had only a small amount of the 1F-PCPDTBT (which was synthesized only for our purpose), and that the remaining solution of the material had actually aged, giving a different device performance. We were however successful in analyzing the less phase-separated PCPDTBT:PCBM blend, and the data have now been added to the manuscript.

Regarding your comment on the PCPDTBT:ICBA blend and its poor morphology, we point out that the two pool model has been mostly applied to analyze data of blends with a high degree of intermixing, because these blends will presumably exhibit the highest density of localized (non-dissociable) CT states.

3) I could not find any details or control experiments on the strong extracting bias. How strong is it and how sensitive is the experiment to the level of bias used? Also the effect of delay time before extraction on the experimental results can give additional insight into which binding energy is actually measured.

The extracting bias was chosen based on the requirement that non-geminate recombination is not a loss channel. The sensitivity of the experiment on the extracting bias partially depends on the field dependent J-V characteristic of the sample, in the 3rd quadrant. Data is shown in Figure SI10, where we measured the jV of a 150 nm thick sample up to saturation (~10 Volts) and compared it to the extracted charge from TDCF. TDCF traces and jV-curve safely overlap at a voltage of -1 V, meaning that a collection bias of -1 V would actually be sufficient to avoid NGR losses at room temperature. However, lowering the temperature requires the collection bias to be more negative, and we identified a collection voltage of -3V to be sufficient (and save) for all of our experiments.

The second part of the question is not entirely clear to us; in this experimental conditions we are measuring only the free and extractable charge; the presence of the long plateau (up to 30 ns) is a confirmation that there is no dispersive recombination, or “cooling” of charges (Figure SI1a, b, e). Otherwise, recombination dynamics would be observed as a faster decay within the mentioned time-window. Additionally, TDCF is not able to resolve a delay shorter than 4 ns due to RC-time limitations. To get closer to what we think your question might be, we added time resolved CT-photoluminescence data in Figure SI1 d and f. As expected the temperature dependence of the CT-exciton lifetime is larger for the PCPDTBT:ICBA than for the PCPDTBT:PCBM device suggesting a lower CT-state binding energy for the latter one. Please see also our extensive reply to the first question of reviewer #2. We defined and discussed the meaning of our measured activation energy more precisely in the main text.

Reviewer #2 (Remarks to the Author):

In the field of organic solar cells there is already for some years an ongoing debate regarding the route of how exciton dissociation occurs : through so-called "hot" CT-states or through thermally relaxed "cold" CT-states. In this manuscript the authors address this challenging issue through the study of the so-called "activation energy for free charge photogeneration" using mainly time-delayed-collection-field (TDCF) experiments. The Neher group is recognized for its expertise in this domain and they have used this technique already in the past to support the view that charge generation in organic solar cells occurs through the relaxed CT states (cfr. reference 26). Of particular interest in this manuscript is the performance of temperature and field dependent experiments yielding the above mentioned "activation energy for free charge photogeneration" which results to be of the order of thermal energy and nearly independent of the excitation energy.

Furthermore, the authors highlight the importance of morphology comparing donor:acceptor blends using fluorinated and non-fluorinated PCPDTBT as donor material and PCBM and ICBA as acceptor material yielding LUMO and HOMO offsets in the same direction but with different effects on morphology.

Although this work is in general solid and of interest for the OPV-community, the following concerns and remarks are formulated:

The main concern regarding this manuscript is that the 'meaning' of the obtained "activation energy" is not sufficiently clarified. In the manuscript a perhaps disproportionate part is devoted to the description of the morphology, while the core issue of the nature, meaning and effects of the obtained "activation energy" could be discussed more extensively. On page 6 the authors state that the activation energy "is a reasonable measure for the binding of the lowest state involved in the pathway of charge generation". On page 21 the authors write "given that our activation energies are independent of excitation energies, such values must represent the effective binding of low lying "thermalized" CT state populations. The formulations regarding the nature of the "activation energy" seem to be insecure and a more extensive description could improve the formulated concern.

Thank you very much for raising these important questions. We have discussed this issue in great detail during preparation of the manuscript. The reviewer is correct in stating that the meaning of the activation energy was not properly discussed in our manuscript. In fact, we might not have properly explained why we studied the temperature dependence of generation and may also have put too much wait on the actual numbers of the activation energy.

Our main goal of this work was to investigate any dependence of the excitation energy on the temperature dependence of free charge generation. This is a well-established way to identify hot generation pathways in neat organic semiconductors. It has been shown that the temperature dependence of free charge generation is very sensitive to the energy landscape involved in the free

charge generation pathways, so if excess energy would alter this pathway, we should see it in the temperature dependent data. Interestingly, that approach has been rarely applied to photovoltaic blends, and we are not aware of a single publication which performed a direct measurement of the generation efficiency as function of temperature over such a wide range in excitation energies, covering direct CT but also hot exciton excitation.

What we observe here is that the activation energy does not depend on the excitation energy. Therefore, our values of the activation energy are a measure of the extra thermal energy which is needed to dissociate “thermalized” CT state populations into free charge, in competition to their geminate recombination. Also, as these activation energies have been derived from zero field generation experiments, barrier lowering through electrostatic effects must be nearly absent. Nevertheless, a direct correlation between the activation energy and the binding energy of the contributing states is still not trivial. In a homogeneous 3D system Braun’s model, predicts the zero field dissociation rate to follow an Arrhenius-type temperature dependence, where the activation energy is exactly the CT binding energy. The same is true for the 1D model developed by Rubel. For poorly performing systems, the temperature dependence of the free generation efficiency tracks the one of the dissociation rate, meaning that under those conditions, the activation energy of the free carrier generation efficiency is also the binding energy. The situation is more complicated in presence of energetic disorder, because now, additional low energy states became available for free charges, resulting in a concurrent reduction of the apparent activation energy. However, the term “binding energy” becomes rather ill defined in highly disordered systems, because free carriers will thermalize well below the maximum of the DOS distribution (and the activation energy might be a better property to describe the underlying energetics). We also like to point out that in our systems, the most ordered system has the lowest activation energy. From that we conclude that disorder is not the main cause of the low activation energies seen here. From this, we conclude that the activation energies measured here are indeed reasonable measures of the energy landscape from the bound relaxed CT state to free charges, and that this energy is significantly reduced by charge delocalization at the heterojunction, microelectrostatic effects, or the stabilization of free charge on the pure donor and/or acceptor.

In order to emphasize our main intentions, we have altered part of the abstract, introduction and conclusion. The new text also includes a short discussion about the meaning and importance of the activation energy.

On page 17 (from line 23) the different activation energies are mentioned but no discussion is provided on why these activation energies are different.

The Reviewer is correct. We have added more comments in the text regarding the reason behind the differences in activation energies, see also above.

The energy of the relaxed CT state is also temperature dependent (cfr. Vandewal et al., Physical Review B, 2010). This effect is seemingly not considered in the present work. How does this affect the proposed results ?

We have also discussed this important point in detail during preparation of the manuscript. A shift of the CT energy, and/or a broadening of the CT band would specially affect our temperature dependent generation data at low photon energies. Indeed, such an effect might be the cause of the stronger decrease of the generation efficiency with decreasing temperature when exciting the PCPDTBT:PCBM blend below the bandgap in Figure 3f. Such an effect is not seen when comparing bias-dependent generation data for different excitation energies. Therefore, there is some evidence that temperature affects the CT onset region, but this effect seems to be small. One possible reason is inhomogeneous disorder. Notably, when carefully analyzing the temperature dependent EQE spectra in the cited paper, the effect of the temperature on the shape of the EQE in the CT region is rather small. Rather than that, decreasing the temperature mainly causes an overall down-shift of the EQE spectrum, pointing to an overall decrease of the generation efficiency independent of photon energy in full agreement to what we observe here.

On page 19 - line 15 - the intrinsic disorder in N2200 is mentioned (33 meV). Why is the energetic disorder not listed in Table 1 - p. 20 ? What is the relation between the energetic disorder and the obtained activation energies ?

Unfortunately, we have no numbers of the energetic disorder in our blends, which would require a detailed investigation of temperature-dependent mobilities. The effect of energetic disorder on the activation energy is addressed in the revised Discussion Chapter, where we point out that energetic disorder has been shown to lower the activation energy of free carrier formation, because it provides additional low energy states for the free carriers.

In relation to the discussed temperature dependence of electron transfer in PCPDTBT:PCBM, we would like to signal to the authors the recent paper by Unger et al. on "The impact of driving force and temperature on the electron transfer in donor-acceptor blend systems" in Journal of Physical Chemistry C - October 2107.

Thank you very much for bringing the article to our attention, which was not published at the time of submission of this paper to Nature Communications in September. The paper by Unger et al. addresses a very different issue, namely the rate of initial charge carrier generation from excited excitons through the photoinduced charge transfer, while our paper questions how incident photon energy, temperature (and field) affects the formation of free carriers. There are several recent papers addressing the rate of charge generation in dependence of energetics and morphology, and we have added an appropriate comment (and references) in the discussion section.

The investigated donor PCPDTBT is not among the current high performance materials. Furthermore, in table SI1 the authors give a list of material systems and their specific issues in relation to the presented TDCF measurement methodology. How general are therefore the obtained results ?

Clearly, more measurements on more systems need to be performed to fully answer this question. However, we believe that the material platform studied here is, indeed, a very well suited model to study how excitation energy affects free charge generation (see also our reply to a similar question of Reviewer 1), but we are keen to extend our studies to more efficient systems in the future.

Why did the authors not insert the IQE curves for the investigated blends ?

Measurements of the IQE have been used in the past to analyze the effect of excitation energy and bias on the efficiency of free charge generation and extraction. However, as pointed out in the introduction of our paper, measuring absolute IQEs is difficult, in particular when aiming at resolving small variations. Often, when IQE is reported, absorption of the active layer and parasitic absorption of the sublayers is not measured in the actual device structure, which leads to errors due to the reflective electrode. Further, measurements of the IQE around the CT energy would require sensitive measurements of the absorption of all layers. Photo-thermal-deflectance-spectroscopy (PDS) would be the appropriate method, but we don't have this method available in our labs and also, it cannot be applied to the full device stack.

We, finally, note that we were particularly interested in the temperature dependence of generation at low internal fields, where the EQE becomes affected by recombination of photogenerated charge with dark-injected background charge, but where the photocurrent might also suffer from trap-assisted recombination, particularly at low temperature. Given the great advantage of TDCF to avoid these issues, we have not considered to determine the temperature and field-dependent IQE on our samples.

Minor remarks :

Thanks a lot for your remarks, we addressed all of them:

Reference 27 and reference 61 are the same.

The references were merged

Consequent format of references.

Done.

Please check carefully the text again - see for instance :

Done.

Page 16 - line 2 : optical bad (=>band) gap

Done.

Supporting information p. 1 - line 21 : Table SI1: Overview of other blends with TDCF and the reason why these blends were NOT (NOT is missing) considered further.

Done.

Supporting information p. 3 - line 12 : the absorption of the neat films is reveals ??

Done.

Supporting information p. 3 - line 18 : the affect (=> effect) of fluorination

Done.

Supporting information p. 3 - line 18 : more pronounced then (=> than)

Done.

Supporting information p. 6 - line 9 : will the surface => while the surface

Done.

Supporting information p. 8 - line 2 : change generation => charge generation

Done.

Y-axis in 3 figures of figure SI8 : => same division

Done.

As mentioned before, this work is in general solid and of interest for the OPV-community, but based on the formulated concerns and questions, we are not convinced that the presented manuscript will 'resolve this debate' ('hot' vs 'cold' CT states) as aimed by the authors in the abstract. Furthermore, we do not consider the presented results and methodology sufficiently groundbreaking (cfr. references 36,40) for publication in Nature Communications, but would recommend publication in a more specialized journal.

We acknowledge the comment however; we do not share the view of the Reviewer. We, specifically note that this is the first time that the generation of free charge in a bulk heterojunction device has been directly addressed over such a wide range in temperature and photon energies. This was only possible though the use of a highly sensitive TDCF setup established in our labs, and the experience which we gained with that method over the years. This allowed us to investigate in great detail how photon energy affects the temperature and electric field dependence of free charge generation. We emphasis here that TDCF (and only TDCF) allowed us to gain access to the temperature dependence of free charge generation in the low field limit where it is not assisted by an electric field, and where it is most sensitive to the energetic landscape. We find that those activation energies are not (within small error bars) affected by the input photon energies, through a range that covers direct CT excitations but also photon energies nearly 1.5 eV above the bandedge. If excess energy would affect

the pathway of free charge generation, we would see it here! We, therefore, firmly believe that our results are important, that they are re-driven from suited systems with a capable method, and that our work is suited for publication in Nature Communication. To emphasis these points, we have partially revised the abstract and also parts of the Introduction.

*In the mentioned publication F. Gao, PRL **114**, 128701 (2015) the authors measured temperature dependent EQE and IQE at an unknown intensity including the difficulties (recombination and extraction limited) stated in the answer of your 7th question as well as the authors did themselves. In addition, the photon energy at which they evaluated the EQE is not given in the paper and probably only one. The second paper M. Gerhard, Phys. Rev. B **95**, (2017) is based on time-resolved photoluminescence experiments. Equation (2) in the paper is $PL = A k_r / (k_r + k_{nr} + k_d)$ in which none of the rates are directly accessible, rendering a conclusion of the dissociation efficiency (goal of the paper) difficult. In TDCF we are able to directly measure the dissociation efficiency η_{diss} which is $\eta_{diss} = k_d / (k_r + k_{nr} + k_d)$. We added the calculations to the SI.*

Clearly, more measurements on more systems have to be performed to fully answer this question. However, we believe that the material platform studied here is, indeed, a very well suited model to study how excitation energy affects free charge generation (see also our reply to a similar question of Reviewer 1), but we are keen to extend our studies to more efficient systems in the future.

We finally want to point out (as we have done above) that the manuscript has been submitted as a Communication, where we apply a dedicated method to a well-selected material system to address an important scientific question. This is not a full article published in one of the main Nature Journals and it's not intended (and suited) to provide universal answers. Taken this into account, we altered some the final statement regarding the generality of our findings. In particular, the statement in the original version that we settled the dispute regarding the contribution by hot generation pathways might have been a bit too strong, given the fact that we did not study a wide range of materials.

Reviewer #3 (Remarks to the Author):

The paper by Kurpiers et al reports on the field dependence of charge photogeneration in polymer-fullerene solar cells, i.e. the field dependent dissociation of (slightly) bound electron-hole pairs at the donor-acceptor interface to extractable charge carriers. The authors state that they

finally settle the dispute on the importance of excess energy (hot generation) on the charge photogeneration from relaxed electron-hole pairs. While I appreciate the experimental evidence provided by the authors and its added value (see below), I believe this claim to be somewhat too strong. In 2010, Lee et al showed this already in a JACS paper, and it was confirmed later by e.g. Vandewal et al in Nature Mater 2014 with contributions from the Neher group. The reported measurements in these publications were close to solar cell operation conditions, in contrast to the measurements by Grancini et al and Bakulin et al which both used fs-

spectroscopy. However, the data shown by Bakulin et al did not, I think, prove that hot generation was dominant, as they even reported its comparatively small relative contribution and highlighted the importance of delocalisation (while saying that higher energy states might be more delocalised). While I would ask the authors of the present paper to state that the verified prior work with conditions close to solar cell operation, instead of claiming to settle a dispute, I am still in favor of publishing the reported research in Nature Communications: The findings on the small (thermal) activation energy by a highly relevant and sensitive technique are very important and relevant to researchers in the field. They can be a reliable basis to describe, e.g. by theoretical work, what finally is the dominant driving force for dissociation of bound pairs (or why the binding energy is so low). Also, the additional study on the potential difference of electroluminescence and photoluminescence in view of the model of two different CT state types is very interesting.

From an experimental perspective, the work is also very relevant to working solar cells due to the low laser fluences used in this study. However, I am a bit concerned about the used laser excitation with different photon energies.

Thanks you for the positive assessment of our work and the helpful comments. It actually turned out to be quite difficult to perform these measurements with the necessary accuracy, and we firmly believe that these data are an important contribution to the ongoing debate on charge carrier generation and possible hot processes. In reply to your comments, we have repeated and extended our PL/EL study, moving this topic more into the focus of the paper.

Please find our reply to all other comments in the following

First, let me point out that this is not the first time this was done, the authors and others also published photon energy dependent data of charge extraction on organic solar cells. I stress that here as the authors state "Surprisingly, we find a weak to negligible effect of the excitation energy on the charge generation activation energy.". As a very weak impact of photon energy on field dependent photogeneration was already reported in literature, replicating this finding (even when using another donor material) should have been not very surprising. Second, the lowest photon energies used in the study, while in the range or even below the CT state energy defined as half way between emission and absorption peak, are still above the relaxed CT energy as given by the electroluminescence emission peak. For instance, the authors use photon energies of laser excitation of 1.17 eV or more, where the relaxed charge carriers reside around 1.1 eV as measured by EL. These are 3 kT difference even at room temperature for the PCPDTBT. It is much more for the fluorinated version which was excited at 1.35 eV, even if the 1.17 eV laser photon energy used before would have been a perfect fit to the relaxed CT energy of (I am estimating) 1.17 eV. This implies that the directly excited bound pairs have an excess energy of several kT and are not perfectly relaxed. This shortcoming is shared with all previous studies applying pulsed lasers, by the way.

The Reviewer is right in saying that we (and others) had performed TDCF experiments in the past, with varying excitation energy. However, with a much more sensitive electronics and a new laser system at hand, our excitation range has been significantly extended, below E_{CT} but also nearly 1.5 eV above the polymer bandgap. Notably, our new study comprises temperature dependent data, which we use as a powerful tool to investigate possible contributions from hot generation pathways. Unfortunately, and as pointed out in the text, excitation with 1064 nm (1.165 eV) only gave signal for the PCPDTBT:PCBM blend with the most red-shifted EQE. For the two other blends, the EQE_PV at 1064 nm was too low to give a well-measurable signal.

We, however, like to point out here that an excitation energy of 1.165 eV, ca. 60 meV above the CT emission maximum at 1.104 eV, does not mean that we excite a state 60 meV above the relaxed CT. This is because the shape of emission spectra is largely determined by the ground state properties. In fact, in the framework developed by Koen Vandewal to describe the absorption and emission of CT states [K. Vandewal et al., *Phys. Rev. B* **81**, 125204 (2010); K. Vandewal et al., *J. Am. Chem. Soc.* **139**, 1699 (2017)], the width of the CT emission is determined given by the slope of the ground state potential at the nuclear coordinate of the fully relaxed CT_1 state, while the emission intensity at a certain transition energy mirrors by the Boltzmann population of the excited CT state (see left graph of Figure 2) For PCPDTBT:PCBM, the EL intensity at 1064 nm is only 15 % lower than at the EL maximum, suggesting that the corresponding CT excited state is within $k_B T$ of the fully relaxed CT exciton (right graph).

Figure 2: Left: Potential diagram of the ground state (GS) and first excited CT state (CT_1) (adapted from K. Vandewal et al., *Phys. Rev. B* **81**, 125204 (2010)). The shape of the CT emission is largely determined by the slope of the ground state potential at the equilibrium nuclear coordinate of the CT_1 state, and related to the reorganization energy. Therefore, excitation around the EL maximum will excite low energy CT states. Right: the EL spectrum of a 110 nm PCPDTBT:PCBM blend. The emission intensity at 1064 nm is 85 % of the intensity at the emission maximum, indicating that it originates from CT states within the thermal population of the thermalized CT manifold.

We have added some more sentences to the corresponding chapter to outline this situation. In addition, we have performed additional experiments using our fs laser source and high sensitive detection, and indeed were able to get a well resolved signal for the PCPDTBT:ICBA blend for 1.3 eV

excitation, which is at the EL maximum of that blend. Unfortunately, we had only a small amount of the 1F-PCPDTBT (which was synthesized only for our purpose), and that the remaining solution of the material had actually aged, giving a different device performance. We are therefore not able to supply TDCF data with excitation at the emission maximum of the 1F-PCPDTBT:PCBM blend.

A note concerning the authors claim that transient absorption cannot distinguish (always) between charges and triplets. That might be so in special cases, but usually they can be separated, as shown for instance by Chow et al, JACS 2014, or the work by Laquai and Howard. Also, the authors state that the sensitivity of TDCF exceeds that of transient absorption. This might also be the case, but I want to point out that transient absorption data over many orders of magnitude in time and charge have been presented, whereas the data in the current manuscript is only shown on a linear scale in terms of charge. I assume the authors refer to the low excitation fluence of their technique, which is indeed a special trademark of the technique and the excellent experimental optimisation of the authors. That said, looking at the charge on a linear scale, I am not sure that dispersive non-geminate recombination can be ruled out, although its importance for the solar cell working conditions likely will be very limited for the shown material system. Also, it is not the focus of this manuscript.

*Firstly, our main concern regarding TAS was that it's not straight forward to differentiate between bound and free charge. A possible way to do so (nicely shown by F. Laquai) is to vary the fluence and to separate geminate and non-geminate recombination, but that requires that measurements can actually be performed over a wide range in carrier densities. The main problem of TAS (also related to that) is the need of high fluences to insure appreciable signal strength, rendering it particularly difficult to perform measurements with direct excitation of the CT state. In fact, there are very few TAS measurements using sub-band gap excitation. One example is B.R. Gautam et al., Adv. Energy Mater. **6**, 1301032 (2016), where blends of PCBM with HTAZ and FTAZ were measured for 2.48 and 1.28 eV excitation. These authors did not observe an effect of excitation energy on charge separation dynamics. However, these blends are quite efficient and a large effect of excitation energy on device performance is anyhow not expected. Also, the authors did not perform measurements as function of temperature. There is TAS work by Laquai and coworkers where they studied the excitation dynamics for different temperatures (e.g. R. Mauer et al., J. Phys. Chem. Lett. **1**, 3500 (2010)), but this work was on P3HT and only for above bandgap excitation. Bakulin employed the pump-push photocurrent scheme to measure different blends as function of temperature, but all these experiments were performed with above bandgap excitation [A.C. Jakowetz et al., J. Am. Chem. Soc. **138**, 11672 (2016)]. An earlier work by Bakulin applied the PPPc method to MDMO-PPV:PCBM with above and below bandgap excitation, where they observed very similar properties. However, they never followed up on that work and no temperature dependent measurements were performed with CT excitation to best of our knowledge. It is yet quite unclear how the PPPc signal is related to free charge generation. There is some nice work by D.H.K. Murthy et al., J. Phys. Chem. C **116**, 9214 (2012) where they used transient microwave conductivity to measure P3HT:PCBM and P3HT:bis-PCBM as function of temperature for below and above bandgap excitations, but this method measures the product of generation efficiency and mobility.*

Here, with TDCF, we investigate directly the temperature- (and field-) dependence of free charge generation, with a hitherto unmatched range of excitation energies. Regarding possible dispersive non-geminate recombination we like to point out that any appreciable dispersive loss would result in

a well-resolved initial slope (see e.g. J. Kurpiers et al., Sci. Rep. 6, 26832 (2016).) which is clearly absent here.

In Fig. 3 f the authors show the field dependence of charge generation. It is unclear to me which voltage bias was used for this cross section of the data, as I cannot identify the "low field plateau region" mentioned in the text.

The question probably refers to the Fig SI7 in the Supporting Information. Here, we plotted the same data as in Figure 3c, but now as a function of electric field. The data reveal that the field-dependence of charge generation becomes considerably weaker when going to low electric fields, as expected. We used the average charge on this low field region to plot the temperature dependent data in Figures 3f and 4c and 4d.

Finally, let me point out again that while I raise some concerns which should be commented upon by the authors, I believe that this study is a very important contribution on charge photogeneration, in particular in view of the very small thermal activation energies; and this for a material system showing a rather pronounced field dependence in contrast to systems such as P3HT or PTB7 blended with fullerene. Also, the careful consideration of the difference between electroluminescence and photoluminescence, reported almost ten years ago, has been verified with high sensitivity under low injection currents to ascertain comparable conditions. In my opinion, this is an important contribution to the discussion around the model with the two CT reservoirs.

Thank you very much for the positive recommendation.

Reviewers' Comments:

Reviewer #1:

Remarks to the Author:

The revised manuscript addresses well most of my (and other reviewers') technical comments. Particularly useful I think is the addition of time-resolved CTPL results acquired in collaboration with Howard team.

The general focus of the paper also improved. And I agree with reviewer #3 that the paper deserves the spotlight provided by Nature Communications.

As minor remark - I would still like to ask the authors to reflect little bit more on my former comments 1-2. It is quite convincing that the amorphous systems they study do dominantly use 'cold' channel for charge separation. However it is still not clear to what class of materials can this conclusion be generalised. I therefore think it would be really valuable for the readers if authors not just present a 'case study' but try to give some hint (even in the abstract or title) what systems would behave similar to those addressed in this study.

Reviewer #2:

Remarks to the Author:

In this revised version of the manuscript the authors have addressed the earlier formulated comments and suggestions in a very thorough manner. The inserted modifications of the title, abstract and of the text body are highly appreciated. This brings the 'story' better to its essence. Furthermore, this version of the manuscript makes it more comprehensible (e.g. interpretation of action energy vs binding energy) and highlights better the true benefits of the used technique and obtained results (e.g. temperature- nor field dependence of free charge generation on excitation energy) which are highly interesting and innovative in the ongoing discussion on the role of CT states in OPV. Based on the formulated arguments, we are now in favor of accepting this revised version for publication in Nature Communications.

A few tiny textual adjustments are needed :

124 : low energy energy

175 => check sentence : recombination 'a' low

430 : temperature-

579 : is needed render

Reviewer #3:

Remarks to the Author:

The authors address my concerns (and, in my opinion, also the points raised by the other referees) adequately. I am still of the opinion that this is important work. I recommend publication without further changes.

Replies to the Reviewers

Thank you very much for your positive feedback to our revised version, but also (once again) for the very detailed reports on the original submission. Your comments provided us with very important input by identifying several weaknesses of the original submission. We are now happy to learn, that you recommend the revised version for publication, ensuring us that we have properly taken into account most of the issues raised in your previous reports.

Enclosed is our reply to your assessment of the actual version of the paper.

Reviewer #1 (Remarks to the Author):

The revised manuscript addresses well most of my (and other reviewers') technical comments. Particularly useful I think is the addition of time-resolved CTPL results acquired in collaboration with Howard team.

The general focus of the paper also improved. And I agree with reviewer #3 that the paper deserves the spotlight provided by Nature Communications.

As minor remark - I would still like to ask the authors to reflect little bit more on my former comments 1-2. It is quite convincing that the amorphous systems they study do dominantly use 'cold' channel for charge separation. However it is still not clear to what class of materials can this conclusion be generalised. I therefore think it would be really valuable for the readers if authors not just present a 'case study' but try to give some hint (even in the abstract or title) what systems would behave similar to those addressed in this study.

The referee is correct in stating that some clear statements are needed regarding the implication from our findings for the understanding and development of other (and more efficient systems) organic solar cells.

Here, we like to point out again that our 1F-PCPDTBT:PCBM blend is quite well performing when considering weak field- and temperature dependence of free charge generation. The reason for the moderate efficiency lies rather in the poor V_{oc} (which is typical for PCPDTBT:PCBM blends) and but also the moderate J_{sc} (mainly because of the rather small layer thickness used here). We therefore consider our series of blends, going from the highly intermixed and amorphous PCPDTBT:ICBA to the well aggregated and phase-separated 1F-PCPDTBT:PCBM system as being useful and relevant for the study of the pathways of generation in relation to excess photon energy and morphology.

Having this in mind, we extended the discussion by two paragraphs stating that important implications of our findings for the understanding and development of OPV devices. We draw attention to the fact that with low lying CTs states always being involved in free carrier generation, the energetics and dynamics of such states becomes of great importance. We cite several examples, where differences in performance of OPV devices were related to the

local packing and orientation of molecules at the DA heterojunction. We further put stronger emphasis on the fact that the formation of pure and well-ordered phases will likely cause a significant reduction of the binding energy of CT states, and cite examples of highly efficient OPV devices which actually exhibit such morphologies. However, we also state that more experiments need to be performed to get a broader picture of the pathways of free charge generation in such high efficiency cells.

Reviewer #2 (Remarks to the Author):

In this revised version of the manuscript the authors have addressed the earlier formulated comments and suggestions in a very thorough manner. The inserted modifications of the title, abstract and of the text body are highly appreciated. This brings the 'story' better to its essence. Furthermore, this version of the manuscript makes it more comprehensible (e.g. interpretation of action energy vs binding energy) and highlights better the true benefits of the used technique and obtained results (e.g. temperature- nor field dependence of free charge generation on excitation energy) which are highly interesting and innovative in the ongoing discussion on the role of CT states in OPV. Based on the formulated arguments, we are now in favor of accepting this revised version for publication in Nature Communications.

A few tiny textual adjustments are needed :

124 : low energy energy

175 => check sentence : recombination 'a' low

430 : temperature-

579 : is needed render

Thank you very much for the positive assessment of the revised manuscript. Your input was very important to rethink some of the interpretations and statements in the previous versions of the paper. The minor textural errors have been corrected

Reviewer #3 (Remarks to the Author):

The authors address my concerns (and, in my opinion, also the points raised by the other referees) adequately. I am still of the opinion that this is important work. I recommend publication without further changes.

Thank you very much for the very positive feedback. As pointed out above your (and the other reviewer's input) was of great help to improve the scientific quality of the manuscript.